# Confinement of excited states in two-dimensional, in-plane, quantum heterostructures

Gwangwoo Kim ⊕[1,2], Benjamin Huet ⊕[3], Christopher E. Stevens ⊕[4,5], Kiyoung Jo[1], Jeng-Yuan Tsai[6], Saiphaneendra Bachu[7], Meghan Leger[7], Seunguk Song ⊕[1], Mahfujur Rahaman ⊕[1], Kyung Yeol Ma[8], Nicholas R. Glavin[9], Hyeon Suk Shin ⊕[10,11], Nasim Alem ⊕[3,7], Qimin Yan ⊕[6], Joshua R. Hendrickson ⊕[4], Joan M. Redwing ⊕[3,7] & Deep Jariwala ⊕[1] ✉

Two-dimensional (2D) semiconductors are promising candidates for optoelectronic application and quantum information processes due to their inherent out-of-plane 2D confinement. In addition, they offer the possibility of achieving low-dimensional in-plane exciton confinement, similar to zero-dimensional quantum dots, with intriguing optical and electronic properties via strain or composition engineering. However, realizing such laterally confined 2D monolayers and systematically controlling size-dependent optical properties remain significant challenges. Here, we report the observation of lateral confinement of excitons in epitaxially grown in-plane $MoSe_2$ quantum dots (~15-60 nm wide) inside a continuous matrix of $WSe_2$ monolayer film via a sequential epitaxial growth process. Various optical spectroscopy techniques reveal the size-dependent exciton confinement in the $MoSe_2$ monolayer quantum dots with exciton blue shift (12-40 meV) at a low temperature as compared to continuous monolayer $MoSe_2$. Finally, single-photon emission ($g^2(0) \sim 0.4$) was also observed from the smallest dots at 1.6 K. Our study opens the door to compositionally engineered, tunable, in-plane quantum light sources in 2D semiconductors.

Exciton confinement in low-dimensional materials modifies the density of states and enhances the Coulomb interaction between electrons and holes, resulting in a range of novel effects for both fundamental physics and device applications. Over the past decade, atomically thin two-dimensional (2D) crystals have been extensively explored in developing quantum optical devices. However, the lack of lateral confinement of excitonic wave functions in such structures has limited their potential for quantum applications. Single photon

[1]Department of Electrical and Systems Engineering, University of Pennsylvania, Philadelphia, PA 19104, USA. [2]Department of Engineering Chemistry, Chungbuk National University, Cheongju 28644, Republic of Korea. [3]2D Crystal Consortium-Materials Innovation Platform, Materials Research Institute, The Pennsylvania State University, University Park, PA 16802, USA. [4]Air Force Research Laboratory, Sensors Directorate, Wright-Patterson Air Force Base, Dayton, OH 45433, USA. [5]KBR Inc, Beavercreek, OH 45431, USA. [6]Department of Physics, Northeastern University, Boston, MA 02115, USA. [7]Department of Materials Science and Engineering, The Pennsylvania State University, University Park, PA 16802, USA. [8]Department of Chemistry, Ulsan National Institute of Science and Technology (UNIST), UNIST-gil 50, Ulsan 44919, Republic of Korea. [9]Air Force Research Laboratory, Materials and Manufacturing Directorate, Wright-Patterson Air Force Base, Dayton, OH 45433, USA. [10]Department of Energy Science and Department of Chemistry, Sungkyunkwan University (SKKU), Suwon 16419, Republic of Korea. [11]Center for 2D Quantum Heterostructures, Institute of Basic Science (IBS), Sungkyunkwan University (SKKU), Suwon 16419, Republic of Korea. ✉e-mail: dmj@seas.upenn.edu

emission from point defects[1–6] and localized strains[7–11] in the 2D crystals have been extensively reported with optically detected magnetic resonance[12–14], such as spin coherence, spin relaxation time, and the nature of spin-spin interactions. However, deterministic positioning of individual defects and achieving uniform emission wavelength have remained a frontier challenge in this field. Several groups have also attempted to solve this challenge of individual exciton confinement in transition metal dichalcogenides (TMDs) monolayers by physically shaping them into quantum dots (QDs) via both top-down[15–19] and bottom-up processes[20–22]. Many such attempts have demonstrated that decreasing the lateral dimensions of the TMD QDs results in a distinct blue shift in both emission and absorption spectra, widely recognized as a signature of quantum confinement[15,19–22]. However, in all cases, these QDs are produced in a manner such that their one-dimensional (1D) edges are exposed. This leads to edge oxidation or covalent chemistry with other functional groups, resulting in energy levels and trap states that affect radiative recombination rates and cause a broader distribution of electronic states. Therefore, achieving a seamless and defect-free interface between 2D QDs and matrix materials within an in-plane 2D combination is a significant milestone that remains to be achieved. Further, even though some evidence of lateral quantum confinement has been observed before, demonstration of single-photon quantum emission from compositionally confined 2D QDs remains unachieved.

In this study, we have successfully demonstrated the lateral confinement of excitons in large area 2D $MoSe_2$ QD@$WSe_2$ matrix heterostructures grown by a metal–organic chemical vapor deposition (MOCVD) method. These heterostructures were created using sequential epitaxial growth to achieve an ultraclean interface. By controlling the reaction time, we can manipulate the size of the triangular $MoSe_2$ QDs in the range of 15–60 nm. Our optical spectroscopic measurements establish size-dependent exciton confinement within the $MoSe_2$ monolayer QDs. Further, our confined heterostructures exhibited quantum emission with ~0.6 nm spectral line width at cryogenic temperatures for dots as small as 10 nm and a single photon purity of $g^2(0) = ~0.4$. Our results serve as an important milestone in achieving quantum confinement and quantum emission in

bottom-up grown 2D QDs at scale opening new avenues for exploring confined excitonic physics and developing novel quantum photonic devices.

## Results

### Growth and characterization of 2D quantum heterostructures

Figure 1a schematically illustrates the growth process for the single-layer $MoSe_2$ quantum dots (QDs) embedded in a single-layer $WSe_2$ matrix by a sequential epitaxial growth using a horizontal MOCVD system. As a first step, the triangular $MoSe_2$ QDs are grown on a c-plane sapphire substrate at a growth temperature of 950 °C for short reaction times (1, 5, 10 min) using $Mo(CO)_6$ metal precursor and $H_2Se$ chalcogen gas (see Methods for details of MOCVD conditions). Note that the size of the $MoSe_2$ QDs can be varied by adjusting the growth time, and this aspect will be discussed further in characterization. After the $MoSe_2$ QD growth, we sequentially grow a single-layer $WSe_2$ matrix around the QDs in the same chamber without taking out the samples. The growth of the $WSe_2$ monolayer is carried out at the same temperature while supplying $W(CO)_6$ metal precursor and keeping the $H_2Se$ throughout the growth to minimize decomposition of the QDs. A strict 2D in-plane growth along the edges of the $MoSe_2$ QDs can be attained in this manner. Before cooling down, the in-plane heterostructure is further exposed to the chalcogen source to heal any vacancies that could have happened during the growth and prevent the formation of vacancies via hydrogen etching, and also to minimize the formation of clusters composed of excess W atoms on the 2D surfaces[23].

The atomic structure of the $MoSe_2$ QDs embedded in the $WSe_2$ matrix was examined using transmission electron microscopy (TEM). Figure 1b shows a low-magnification annual dark-field scanning TEM (ADF-STEM) image of the in-plane $MoSe_2$ QDs@$WSe_2$ heterostructures after the film was detached from the sapphire substrate and transferred to a TEM grid (see Methods for details). The presence of the $MoSe_2$ QDs embedded in the $WSe_2$ matrix is confirmed owing to the $Z$ contrast mechanism of the ADF-STEM imaging technique[24], wherein the $MoSe_2$ QDs appear darker compared to the $WSe_2$ matrix. It is worth noting that the size of the $MoSe_2$ QDs can be reduced to 15-60 nm by

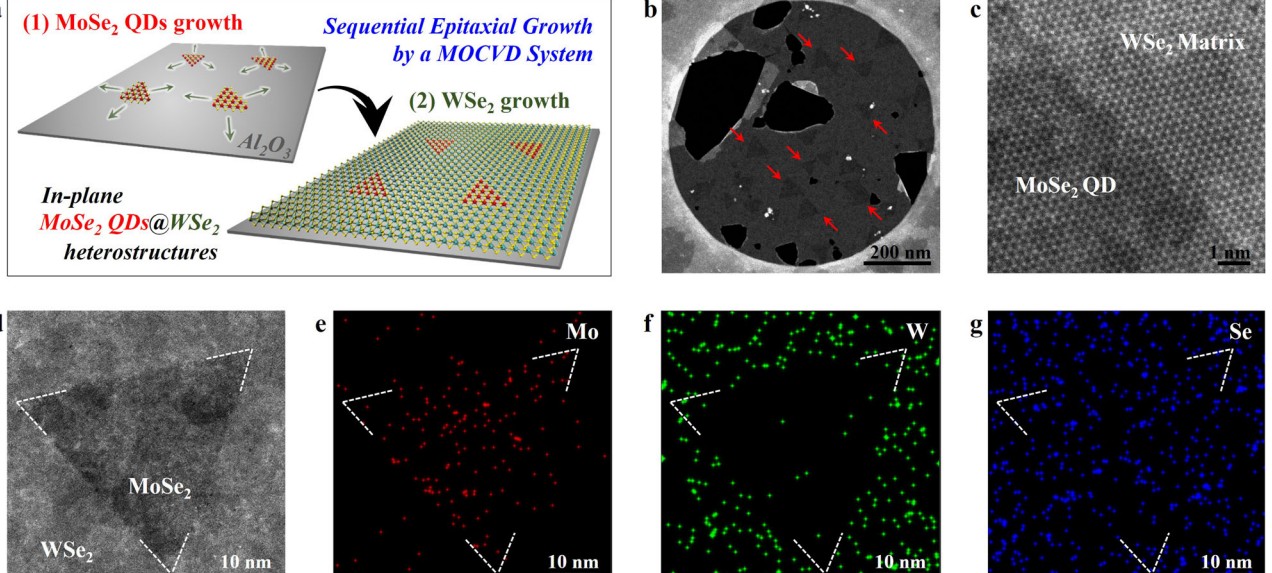

**Fig. 1 | Sequential epitaxial growth of the in-plane $MoSe_2$ QDs@$WSe_2$.**
**a** Schematic representation of sequential epitaxial growth of in-plane QDs heterostructures via MOCVD. **b** Low magnification ADF-STEM image of the heterostructures consisting of the 5 min-$MoSe_2$ QDs and the $WSe_2$ matrix. The $MoSe_2$ QDs are marked with the red arrows. **c** Atomic-resolution ADF-STEM image showing an interface between the $MoSe_2$ QDs and the $WSe_2$ matrix (bottom left and top right, respectively). **d** ADF-STEM image of the heterostructures with the 5 min-$MoSe_2$ QDs. **e–g** Corresponding STEM-EDS element maps of Mo (**e**), W (**f**), and Se atoms (**g**) for the Figure (**d**). The boundaries between the $MoSe_2$ QD and the $WSe_2$ matrix are marked with a white dot line from Figure (**d**).

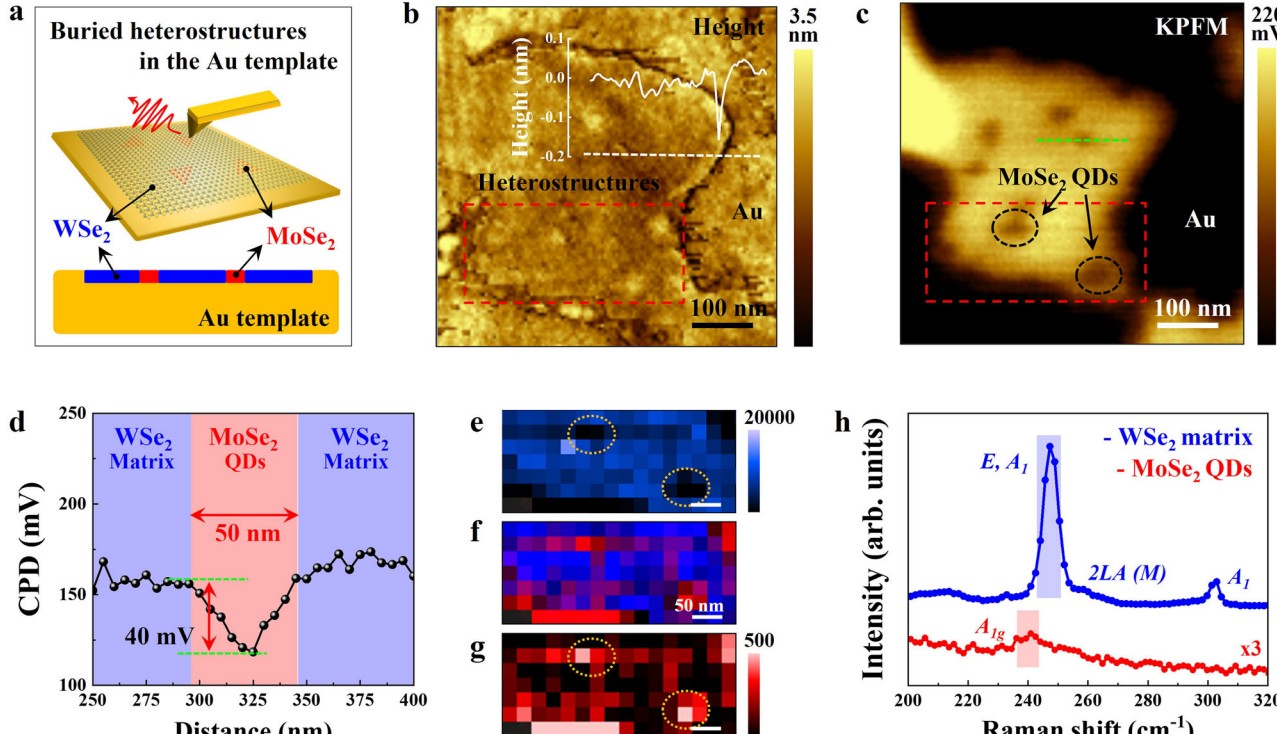

**Fig. 2 | Nanoscale optical and electrical imaging of the MoSe₂ QDs embedded in the WSe₂ matrix.** **a** Schematic representation of nanoscale scanning probe techniques (top) on the 5 min-MoSe₂ QDs@WSe₂ heterostructures buried in an Au template (bottom: cross-sectional view). The samples are prepared by Au-assisted transfer process[25,26]. **b, c** AFM height image and surface potential map of the heterostructures consisting of the 50 nm-sized MoSe₂ QDs and the WSe₂ matrix. (Inset) The height profile marked with a white line in (**b**) shows no discernible height difference across the crystal, proving truly in-plane epitaxy (embedding) of the MoSe₂ QD in the WSe₂ matrix. The MoSe₂ QDs are marked with black dotted circles. **d** Surface potential profile following the green dotted line in (**c**). **e–g** TERS spatial maps of the quantum heterostructures following the red dotted squares in (**b**) and (**c**). The TERS images were created within the spectral ranges of 245–255 cm⁻¹ (**e** WSe₂ A₁ mode) and 235–245 cm⁻¹ (**g** MoSe₂ A₁g mode) with a step size of 25 nm. The MoSe₂ QDs are marked with yellow dotted circles. **f** Overlaid image of (**e**) and (**g**). **h** TERS spectra of the MoSe₂ QD regions (red) and the surrounding WSe₂ matrix (blue) as highlighted in the TERS map.

controlling growth time (Supplementary Fig. S1). Moreover, an atomic-resolution ADF-STEM image (Fig. 1c and Supplementary Fig. S2) obtained from the interface illustrates that the atomically sharp interface formed between the QD and the matrix ensures the in-plane heteroepitaxy of the 2D heterostructures. This is further supported by the clear spatial separation of the Mo and W signals in the energy dispersive spectroscopy (STEM-EDS) maps (Fig. 1d–g).

In addition, the formation of the MoSe₂ QDs is confirmed using micro-Raman spectroscopy with a 633 nm laser on the heterostructures transferred onto SiO₂ substrates (Supplementary Fig. S3). The Raman spectra of the in-plane quantum heterostructures exhibited typical Raman signals of MoSe₂ and WSe₂, including the characteristic A₁g symmetry Raman modes of MoSe₂ (red) and WSe₂ (blue) at 242 and 250 cm⁻¹, as well as WSe₂ resonance peak (green) with 633 nm laser at 239 cm⁻¹. All Raman modes showed no change in the peak position depending on the QDs growth time, but the intensity of both A₁g peaks varied. Supplementary Fig. S3c depicts the plot of the A₁g intensity ratio on the MoSe₂ QDs@WSe₂ matrix as a function of the growth time, which indicates a substantial drop in the intensity ratio with the QDs population. These findings are consistent with the increase in the average size and density of the QDs as the growth time is prolonged.

## Nano-optical imaging of the quantum heterostructures

The far-field Raman spectroscopic results described above sample an ensemble of QDs in the WSe₂ matrix since the laser spot size (~1 μm) is much larger than the average diameters or spacing between the QDs. Therefore, it is difficult to display the actual QD size and distribution

via purely optical means because of the resolution limitations of the instrument. To determine the spatial position and study the band alignment between the MoSe₂ QDs and the WSe₂ matrix, we perform tip-enhanced Raman spectroscopy (TERS) combined with Kelvin probe force microscopy (KPFM) on the heterostructures. First, an epoxy-assisted template stripping procedure is used to prepare the 5min-MoSe₂ QDs@WSe₂ heterostructures buried in a metallic Au substrate (Fig. 2a, see Methods in details) which enhances optical signals due to the high local electric field and the Purcell effect for Raman scattering phenomena[25,26]. As shown in the atomic force microscopy (AFM) height image (Fig. 2b), despite a slight interface gap between the heterostructures and the Au template, they displayed a smooth topography with minimal surface roughness (Heterostructure: 0.09 nm, Au: 0.13 nm). This suggests that the flat sample prepared by the Au stripping process offers more spatially uniform contact. KPFM mapping of the quantum heterostructures in Fig. 2c exhibits a noticeable contrast in the contact potential difference (CPD) between the MoSe₂ QDs (dark, marked with black dotted circles) and the surrounding WSe₂ matrix (bright). This potential difference (~40 mV) arises due to the disparity in work functions between the two materials[27]. Although we measure a smaller difference than expected (~190 meV) due to the quantum-confined dots structure, the difference in potential allows the two materials to be clearly distinguished. The built-in potential of the quantum heterostructure will drive the electron diffusion from the WSe₂ matrix to the MoSe₂ QDs along the direction of the built-in field, showing its typical two semiconductor-based heterostructure characteristic[28,29]. Additionally, it is noted that the size of the MoSe₂ QDs embedded in the WSe₂ matrix is found to be

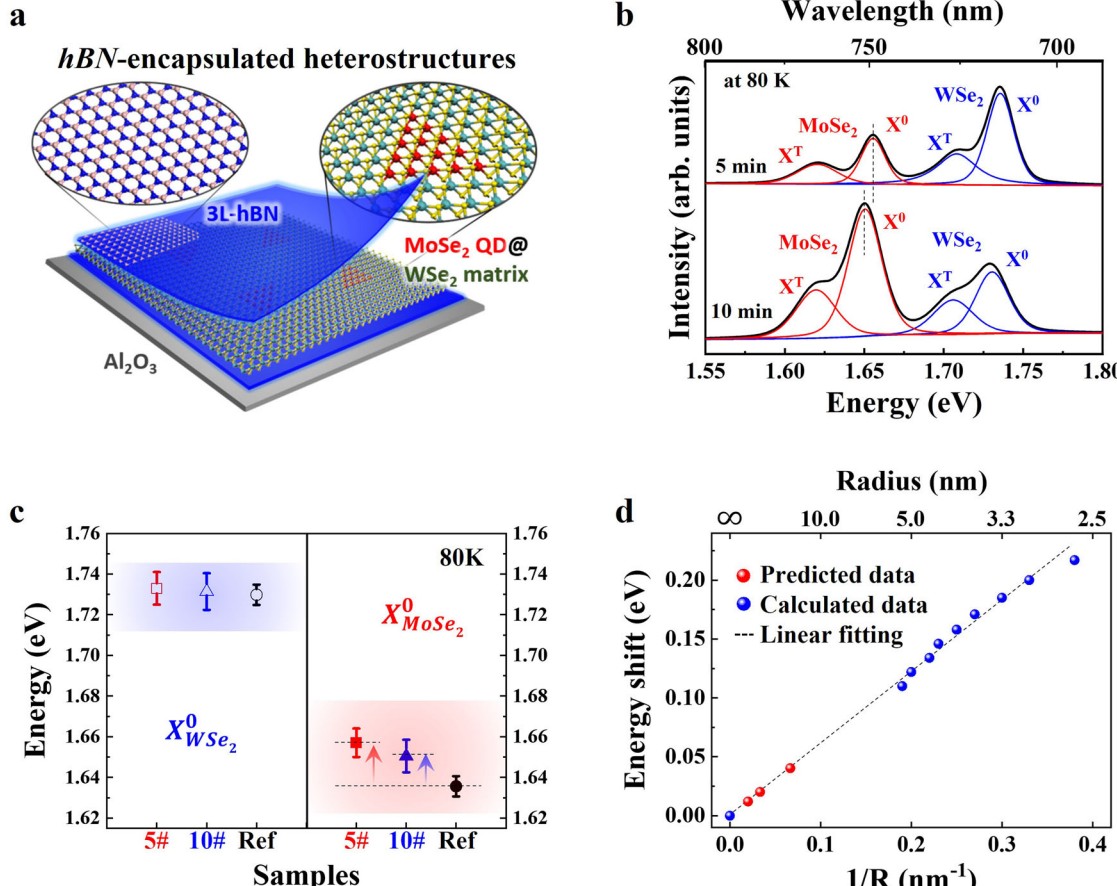

**Fig. 3 | Exciton confinement of the MoSe$_2$ QDs embedded in the WSe$_2$ matrix.**
**a** Schematic representation of the in-plane MoSe$_2$ QDs@WSe$_2$ heterostructures encapsulated in top and bottom h-BN tri-layers. The Mo, W, Se, B, and N atoms are represented in red, green, yellow, pink, and blue, respectively. **b** PL spectra of the heterostructures with different growth times of the MoSe$_2$ QDs (top: 5 min, bottom: 10 min) measured at 80 K. These spectra were obtained by a 633 nm CW laser with an excitation power of 20 µW and a 50× lens with 0.35 NA. **c** Comparison of PL energy position of the main neutral excitons of MoSe$_2$ and WSe$_2$ in the heterostructure samples (red: 5 min, blue: 10 min) with reference monolayers (Ref., black). The points are plotted from the peak positions in (**b**). The reference MoSe$_2$ and WSe$_2$ monolayers are prepared in the same MOCVD chamber. The PL energy positions of the main neutral excitons of WSe$_2$ are 1.733 ± 0.008 (5 min), 1.731 ± 0.009 (10 min), and 1.730 ± 0.005 eV (Ref), respectively. The PL energy positions of the main neutral excitons of MoSe$_2$ are 1.652 ± 0.007 (5 min), 1.651 ± 0.008 (10 min), and 1.636 ± 0.005 (Ref), respectively. **d** Relation between the energy shift and the size of the MoSe$_2$ QDs embedded in the WSe$_2$ matrix. The dashed blue line represents the linear fit of the energy shift with respect to the inverse size of QDs. The blue dots represent the computationally estimated energy shift in the QDs, with sizes ranging from 2.7 to 5.3 nm. The three red dots are predicted energy shifts in 15, 30, and 50-nm quantum dots using linear extrapolation. The experimentally determined optical band gap of pristine MoSe$_2$ is set as zero energy and associated with an infinitely large ($R = \infty$) quantum dot.

50 nm (Fig. 2d), which is consistent with the size observed from the TEM measurement (Fig. 1b). Despite our efforts to establish a clean physical interface (Fig. 1c and Supplementary Fig. S2), the expected sharp potential drop is not observed due to the limitations of our scanning probe system and Au tip diameter of ~20 nm. TERS spectra are collected from the MoSe$_2$ QDs (red) and the WSe$_2$ matrix (blue) regions in Fig. 2e–g. Unlike the previous far-field measurement using a 633 nm laser, a 785 nm laser used for gap mode TERS measurement between the tip and the underlying Au template was weakly resonant with both materials. The TERS spectra from the MoSe$_2$ QD and the WSe$_2$ matrix exhibited several features similar to resonant far-field Raman peaks (Supplementary Fig. S3). Still, we analyze the out-of-plane A$_{1g}$ mode, which was prominent by signal enhancement in the gap mode. Interestingly, the corresponding TERS maps in the ranges of 300-307 cm$^{-1}$ (WSe$_2$ A$_1$ mode, Fig. 2e) and 240–244 cm$^{-1}$ (MoSe$_2$ A$_{1g}$ mode[30], Fig. 2g) allowed for the identification of the 50 nm-sized MoSe$_2$ QDs embedded in the WSe$_2$ matrix. By using different color channels to render the intensity for the WSe$_2$ (blue) and the MoSe$_2$ QDs (red), a color-integrated image (Fig. 2f) shows the spatial composition distribution in these in-plane quantum heterostructures. Furthermore,

a scanning tunneling microscopy/spectroscopy (STM/STS) study to provide more detailed information on the interface and defect states would be valuable for this material system and present an opportunity for future research. Such a study, while challenging on an insulating growth substrate, could be done with improved device/sample architectures[31–33] and sample preparation techniques that minimize or eliminate polymer contaminations[33]. Nonetheless, our TEM results (Fig. 1, Supplementary Figs. S1 and S2) provide sufficient evidence of the abruptness of the quantum heterostructure at the atomic level.

**Exciton confinement in the quantum heterostructures**
Thin hexagonal boron nitride (h-BN) films (Supplementary Fig. S4) are employed to encapsulate the top and bottom of the quantum heterostructures, as shown in Fig. 3a, creating stable structures with confined excitons and protecting the samples from unwanted contamination during the measurements. The large area-h-BN tri-layers are grown on a c-plane sapphire substrate by a CVD method[34]. These tri-layers are then stacked vertically with the in-plane heterostructures, using a polymethyl methacrylate (PMMA)-assisted wet-transfer technique on a SiO$_2$ substrate (see the detailed process in Supplementary

Fig. S5). Note that, in the transfer process, we modified the order of a layer-by-layer stacking to minimize the PMMA residue between the layers. The PMMA-coated top h-BN layers was first transferred to the as-grown quantum heterostructures on the sapphire substrate, followed by coating the samples with PMMA again and then transferring them onto the bottom h-BN film. This method allowed residues to be left on the h-BN while ensuring that clean interfaces were established without residues on either side of the quantum heterostructures. Finally, the stacked samples are transferred onto the $SiO_2$ substrate and annealed at 300 °C with Ar flow (50 sccm) in a vacuum tube furnace. This facilitated better contact by eliminating any trapped solvent or gas molecules at the interface between each layer.

The confinement of excitons in the encapsulated quantum heterostructures is examined using micro-photoluminescence (PL) spectroscopy with a 633 nm-excitation laser. Supplementary Fig. S6 shows the temperature-dependent PL spectra of the $MoSe_2$ QDs@$WSe_2$ quantum heterostructures encapsulated in h-BN films, where the $MoSe_2$ QDs are grown for 5 min and 10 min. The PL peaks show a considerable shift to higher energies with decreasing temperature due to the lattice shrinkage and reduced electron-phonon interaction at low temperatures[35]. At room temperature (red spectra in Supplementary Fig. S6), the PL intensities of the $MoSe_2$ QDs on both samples are not large due to the small size and low density of the QDs. However, at ~80 K, the emission signal from the QDs signal is clear as compared to that of the $WSe_2$ matrix, shown in Fig. 3b. The PL intensities ($X^0 + X^T$) ratio of $MoSe_2$ to $WSe_2$ is plotted in Supplementary Fig. S7, showing strong enhancement at low temperature, possibly due to dominant electron transfer from the $WSe_2$ matrix to the $MoSe_2$ QDs by the band structure modulation[36]. Furthermore, the confined excitons in the heterostructures can be modulated through electrostatic gating. We fabricate a gate-tunable device using mechanically exfoliated graphene and h-BN flakes as the top-gate electrode and the dielectric, respectively. By adjusting the Fermi level in two materials with Type II band alignment, we further demonstrate via emission spectroscopy that efficient electron transfer can be controlled via gating in these in-plane quantum heterostructures (see details in Supplementary Fig. S8).

The quantum confinement of excitons in the $MoSe_2$ QDs is also confirmed by comparing their PL energy positions with those of large-area $MoSe_2$ and $WSe_2$ monolayers grown on sapphire substrates in the same MOCVD system. The PL energy position of the neutral exciton of $MoSe_2$ (right) and $WSe_2$ (left) on the quantum heterostructures ($MoSe_2$ QD growth time of 5 min: red, 10 min: blue) and reference monolayers ($MoSe_2$ and $WSe_2$: black) are compared as shown in Fig. 3d. Depending on the samples, there is no change in the $WSe_2$ PL position. Still, the excitonic features of the $MoSe_2$ PL were blue-shifted on the QD samples, indicating the exciton confinement. The confinement effect is larger in the 5 min-$MoSe_2$ QDs heterostructures (~20 meV) than in the 10 min sample (~15 meV), indicating a stronger blue shift with a smaller lateral size of the QDs. It is worth noting that although the QDs lateral size is larger than the exciton Bohr radius (~1.5 nm) in the TMD monolayers[37,38], the excitons can still be in a weak lateral confinement regime[15,19–22]. Theoretical calculations are discussed below to explore this confinement effect further.

We perform first-principle calculations based on density functional theory (DFT) to investigate the shift of optical transitions due to the confinement effects in QDs with various sizes. Equilateral triangular $MoSe_2$ quantum dots embedded in $WSe_2$ with edge lengths ranging from 2.7 to 5.3 nm were studied. As the size of the QDs decreases, the absolute energy of the conduction band minimum (CBM) rises due to the quantum confinement effect, while that of the valence band maximum (VBM) decreases at a much slower rate. The energy difference between the CBM and the VBM increases with increasing dot size (Supplementary Fig. S9). For instance, in the 2.7-nm quantum dot, the projected density of states reveals that the CBM is

mainly contributed by the quantum dot, with the corresponding wave function being spatially localized within the dot (Supplementary Fig. S10). Note that the electronic state at the VBM is delocalized, while the state at 0.25 eV below the VBM turns out to be confined by the quantum dot. This observation is consistent with the type-II band alignment between $MoSe_2$ and $WSe_2$, which effectively forms quantum dots for electrons but does not imply confinement for those holes at the VBM.

Next, we evaluate the effect of quantum confinement on the intra-dot optical transition by defining an energy shift between two relevant energies: (1) the energy difference between the CBM of QDs and the VBM of pristine $MoSe_2$; (2) the optical gap of pristine $MoSe_2$. To account for the excitonic effects and the underestimation of band gaps by DFT, we increase the calculated band energy differences by 0.11 eV (which is the energy difference between the calculated band gap and the measured optical gap of $MoSe_2$) to roughly evaluate the optical transition energies[39]. The energy shift is plotted as a function of the inverse size of QDs. Due to the limitations in computational capacity, linear extrapolation is used to estimate the results for more sizable QDs. The energy shift is estimated to be 40, 20, and 12 meV for 15, 30, and 50-nm quantum dots, respectively (Fig. 3d). The predicted shift of transition energies in 30- and 50-nm quantum dots are comparable with the observed energy shifts of 21 and 15 meV in the 5 min- and 10 min-$MoSe_2$ QDs (Fig. 3c). It is noted that the discussion of results pertaining to the 1 min-$MoSe_2$ QDs will be addressed in more detail later. Although our calculations do not explicitly include the excitonic effects, which can be significant in two-dimensional materials, the consistent results suggest that the change in the band energies of those electronic states involved in optical transitions may be dominant in the observed shift of transition energies. This finding is consistent with the conclusions of previously reported theoretical[19,40] and experimental results[15,19,41–43] in the 2D TMD QDs. Note that the effects of quantum dot confinement on electrons and excitons are not completely equivalent. Based on the Bethe–Salpeter Equation, the first-order effect of the quantum dot size on excitonic energies involves the shifting of electron and hole state energies. Furthermore, the size of the quantum well may affect the excitonic interactions among these states and thus induce further shifts or broadening of excitonic peaks. Additionally, it is important to note that the exciton–phonon interactions may influence both the decrease in exciton peak energy and the line width, while our computational results do not consider the effect of temperature. Nevertheless, it is expected that temperature effects play a minimal role in the relative shift of the excitonic peaks concerning quantum dot size, which is the primary focus of this computational study.

In addition to PL analysis, the confinement effect in the quantum heterostructures is also verified through reflectance measurements (Supplementary Fig. S11). Supplementary Fig. S11a displays the reflectance spectra of the 10 min-$MoSe_2$ QDs heterostructures (black) and reference monolayers ($MoSe_2$: red, $WSe_2$: blue) obtained at ~80 K. As observed in the PL results, the confinement of the $MoSe_2$ QDs is evident in the reflectance spectra, while the positions of $WSe_2$ absorption peaks are nearly identical between the heterostructures and the $WSe_2$ monolayer. These two optical measurements provide clear proof of the optical confinement effect.

In our previous analysis of temperature-dependent PL, a stronger confinement effect was expected on the heterostructures of the 1 min growth time-$MoSe_2$ QDs with smaller sizes (approximately 15 nm). However, due to the limited size and density of the quantum dots, it is challenging to detect the $MoSe_2$ PL signal (Supplementary Fig. S12). This is mainly because the confocal Raman system, with a nitrogen-cooling stage with a large beam size from a 50×, 0.35 NA lens, is not able to probe and detect the small dots effectively. Therefore, to address this limitation and directly explore the quantum confinement on the QD heterostructures, we perform cryogenic PL measurements

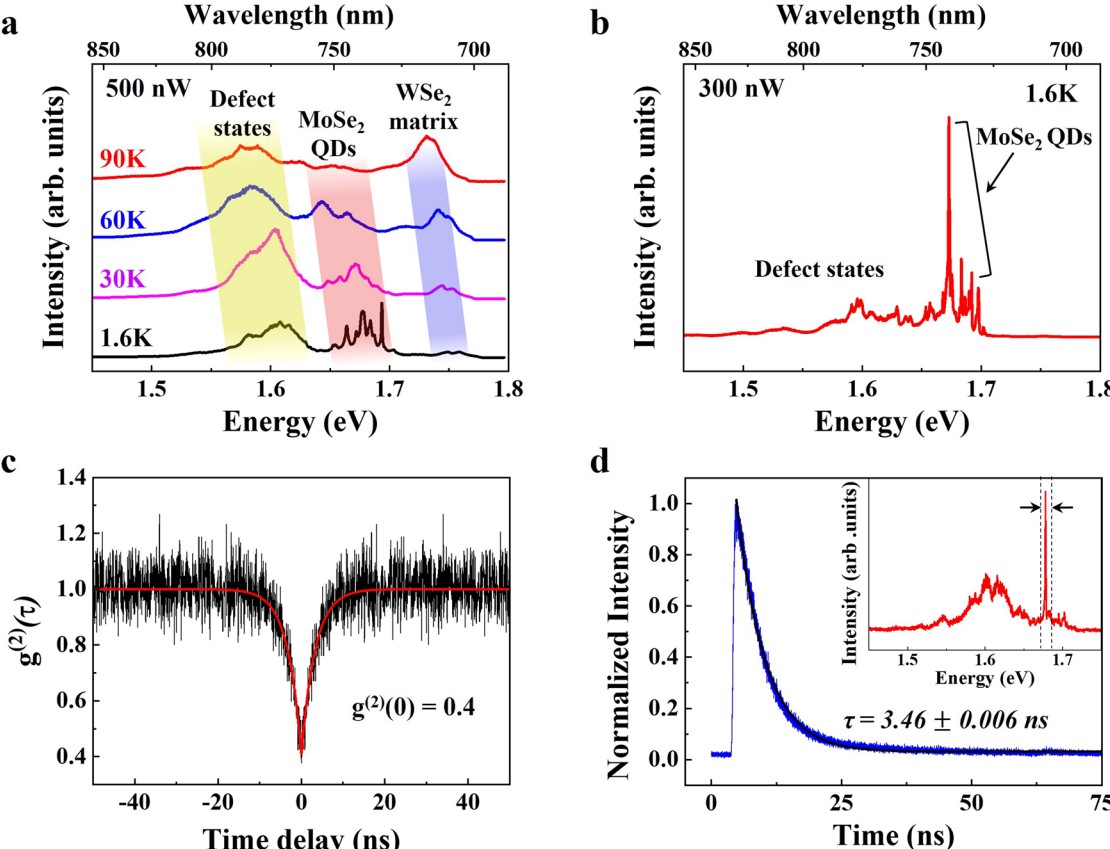

**Fig. 4 | Cryogenic PL measurement and single photon emission in the quantum heterostructures. a** Temperature-dependent PL spectra (90 K to 1.6 K) of the heterostructures with the 1 min growth time-MoSe$_2$ QDs, measured using a pulsed laser excitation (500 nW, 640 nm, 1 MHz). The emission peaks from defect states, MoSe$_2$ QDs, and WSe$_2$ matrix are marked with yellow, red, and blue regions, respectively. **b** Representative cryogenic PL spectrum of the QD heterostructures with an excitation power of 100 nW at 1.6 K. **c** Second-order photon correlation curve for the PL signal of the MoSe$_2$ QDs. **d** TRPL spectrum for the sharp emission of the MoSe$_2$ QDs (1.688 eV), shown in (**b**). The time-resolved PL data (blue line) are convoluted (black line) with the instrument response function, using an exponential function $I = A \cdot \exp(-t/\tau)$. TRPL spectrum for the defect states (1.60 eV) was added to Supplementary Fig. S17.

(down to 1.6 K) with a 100× lens with 0.82 NA. Figure 4a lists the temperature-dependent PL characteristics of the 1 min-MoSe$_2$ QDs sample, showing a clear MoSe$_2$ PL signal. A gradual blue-shift of the peaks (MoSe$_2$ QD, WSe$_2$, and defect states) with decreasing temperature is clearly observed, as in the previously shown Supplementary Fig. S6.

Additionally, we also observe a distinct and sharp MoSe$_2$ PL emission when the temperature approached 1.6 K. We hypothesize that they originate from the quantum-confined main excitons of the MoSe$_2$ QDs because of the matching energy values as discussed above. Figure 4b shows the PL spectra on the quantum hetero-structures (1 min growth time-MoSe$_2$), which are obtained with a pulsed excitation laser (640 nm, 1 MHz) with 100 nW power at 1.6 K showing an emission with a narrow line width (612 ± 124 µeV) at an energy of 1.675–1.692 eV. However, this narrow emission also resembles single photon emitters from dark excitons and defect states in WSe$_2$ monolayers, as reported in several prior studies[1,2,7,11]. In fact, although we perform a chalcogen supply during cooling to minimize defects and vacancies, healing them completely is challenging[44]. Additionally, the formation of Mo-based WSe$_2$ defects introduced by residual Mo sources from the MoSe$_2$ growth process, or 2D TMD defects possibly formed during the transfer process, cannot be disregarded. To eliminate the possibility for these narrow emission lines to be attributed to defect states of the WSe$_2$ matrix, we prepared and measured MOCVD-grown pure 2D WSe$_2$ and MoSe$_2$ monolayers in the same chamber with identical growth conditions.

As shown in Supplementary Fig. S13b, the emission from the defect state of WSe$_2$ was observed at a slightly lower energy position (1.65–1.66 eV), and is significantly broad, as compared to the quantum heterostructures. Further, the main neutral exciton in pure MoSe$_2$ samples was observed at 1.652 eV (Supplementary Fig. S13b, bottom). It is worth noting that the quantum emission observed on the heterostructures (Fig. 4d, inset) was blue-shifted by ~40 meV due to the confined excitons in the quantum dots, which is consistent with the theoretically predicted value in Fig. 3d. Furthermore, to clarify the quantum emission from the MoSe$_2$ QDs, we prepared control samples comprising only MoSe$_2$ QDs with different growth times without the WSe$_2$ matrix (Supplementary Fig. S14) and performed cryogenic PL measurement on the samples at 1.6 K (Supplementary Fig. S15). The 1-min MoSe$_2$ QDs (red line) showed sharp emission at ~1.7 eV, indicative of quantum-confined excitons. However, only broad defect state emissions are observed in the 5- and 10-min samples (blue and black lines). This observation indicates that QDs with smaller sizes are less impacted by edge defect states, which otherwise induce classical emission in larger QDs. Note that the intensity of quantum emission didn't reach the levels seen in MoSe$_2$ QD/WSe$_2$ heterostructures. This observation emphasizes the importance of edge passivation. Our findings illustrate that in quantum heterostructures, light absorption in the WSe$_2$ matrix facilitates exciton confinement within the MoSe$_2$ QDs, enhancing light emission and providing insights into the interaction between material interfaces and quantum confinement. This also suggests and merits future

work on a detailed understanding of the effect of QD size and edge chemistry on quantum emission from as-grown MoSe₂ QDs.

To further validate that the emission acts as a truly quantum light source, we measure the second-order correlation function $g^2(t)$ using a Hanbury–Brown–Twiss (HBT) set-up with two single photon-counting avalanche photodiodes (APDs). Figure 4c shows the second-order correlation under continuous-wave (CW) excitation of the emitter on one of the selected sharp emission lines binned in between the dotted lines (Fig. 4d, inset) using a broad band-pass filter and tunable short and long pass filters. By fitting the measured data with a standard two-level antibunching function, we calculate a $g^2(0) = 0.4 \pm 0.02$, which drops below the threshold for a single quantum emitter of 0.5 (more spectra on several other positions on the sample are provided in Supplementary Fig. S16). This confirms that the MoSe₂ QD is truly a single photon emitter. Figure 4d shows the PL lifetime of the MoSe₂ QDs on the heterostructures, which was measured by excitation with a 640 nm, ~200 ps, 10 MHz pulsed diode laser and sending the spectrally filtered output around the quantum dot wavelength to a single-photon-counting APD. The measured long lifetime of ~3 ns from the exponential fitting (black line) is consistent with the behavior shown by typical III–V semiconductor quantum dots[45–48].

## Discussion

In conclusion, we have successfully demonstrated the lateral confinement of excitons via compositionally controlled in-plane 2D quantum heterostructures of the MoSe₂ QDs embedded in the WSe₂ matrix. These heterostructures show quantum-confined emission that is significantly blue-shifted from the main neutral excitons of pure 2D monolayer MoSe₂. Further, the wavelength and intensity of the emission can be modulated passively according to the QD size and actively via electrostatic gating since the QDs are embedded in the WSe₂ matrix. Our work represents a significant step toward the synthetic control of truly two-dimensional in-plane epitaxial QDs, making them a versatile and tunable quantum light source. To fully exploit their potential for future, high-fidelity quantum light sources, further work must focus on their controllability in terms of spatial position, density, and composition.

## Methods

### Growth of the in-plane MoSe₂ QDs@WSe₂ heterostructures

The in-plane MoSe₂ QDs@WSe₂ heterostructures are synthesized on sapphire substrates in a horizontal MOCVD reactor[49]. The process involved introducing Mo(CO)₆ or W(CO)₆ metal precursors and an H₂Se chalcogen source at a growth temperature of 950 °C. After the heating ramp, a 10 min high-temperature annealing step is carried out under a pure H₂ atmosphere to remove surface impurities and stabilize the sapphire surface. The growth of the MoSe₂ QDs was initiated by introducing Mo(CO)₆ metal precursor ($6.1 \times 10^{-3}$ sccm) and H₂Se chalcogen source (200 sccm) simultaneously. Pure H₂ was used as a carrier gas to transport the metal carbonyl in the chambers via a bubbler which was maintained at a controlled pressure of 760 Torr and temperature of 20 °C. The desired size of the MoSe₂ QDs (10–50 nm) is typically adjusted by the growth time (1–10 min). Sequentially, stopping the Mo precursor supply, W(CO)₆ precursor ($8.7 \times 10^{-4}$ sccm) is introduced to grow the WSe₂ matrix around the MoSe₂ QDs. During the growth of the heterostructures, the identical flow of H₂Se gas continued to be supplied, and the system pressure was maintained at 200 Torr. Finally, after the growth, the furnace is cooled to 300 °C in a mix of H₂/H₂Se and is then cooled further to room temperature in N₂.

### Preparation of buried heterostructure in the Au templates

A thin gold film (100 nm) is deposited onto the surface of the in-plane MoSe₂ QDs@WSe₂ heterostructures grown on the sapphire substrate by using an e-beam evaporator (Kurt J. Lesker PVD-75) under a high vacuum. Then, a Si wafer is attached to the outer gold surface using an epoxy resin. Once the epoxy is cured at 80 °C for 2 h, the gold–sapphire interface is separated by peeling. The in-plane heterostructures are more strongly bound to the Au film and are thus separated from the sapphire substrate. The process results in the transfer of the heterostructures from the sapphire surface to being inlaid in the gold film, exposing the pristine surfaces of the heterostructures that were previously in contact with the sapphire substrate.

### Device fabrication for electrostatic gating

For electrostatic gating device preparation, the mechanically exfoliated h-BN and graphene layers are transferred over the sample by using a polydimethylsiloxane (PDMS)-based dry transfer process to use them as each a dielectric and a top gate electrode. Next, the fabrication of Ti (10 nm)/Au (100 nm) electrode contacts is achieved by using electron beam lithography (Elionix ELS-7500EX) and the e-beam evaporator (Kurt J. Lesker PVD-75). Finally, the samples are cleaned in acetone for the lift-off process.

### Optical and structural characterization

Far-field Raman and PL spectroscopy are performed in a Horiba Lab-Ram HR Evolution confocal microscope with 633 nm excitation lasers. The signals are collected through a 50× microscope objective (Olympus SLMPLN, NA = 0.35) for low-temperature measurements (from room temperature to 80 K). Also, for the low-temperature analysis, samples are placed in a Linkam stage with a liquid nitrogen supply while cooling and heating and pumped to $5 \times 10^{-3}$ Torr during the measurement. Additionally, for electrostatic gating, the electrical bias is applied using a Keithley 2450 sourcemeter. An OmegaScope Smart SPM (AIST-NT) setup is used for topography scans. For tip-enhanced Raman measurements, Au-coated OMNI-TERS probes (APP Nano) are used in the identical AFM setup coupled to a far-field Horiba confocal microscope with a 785 nm excitation laser.

For the cryogenic PL (from 80 K to 1.6 K), time-resolved PL, and $g^{(2)}(\tau)$ measurements, the sample is placed in a cryostat with an in situ 0.82 NA 100× objective. The excitation spot size was approximately 1 μm. For time-resolved PL, the sample is illuminated with 640 nm, 200 ps light generated from a PicoQuant diode laser. For PL saturation and $g^{(2)}(\tau)$ measurements, the sample is illuminated with a 640 nm CW diode laser. The signal is collected in a reflection geometry and routed to a spectrometer for PL measurements. For time-resolved PL and $g^{(2)}(\tau)$, the SPEs are first identified using PL and then spectrally filtered with angle-tunable Semrock filters before being sent to two fiber-coupled Si avalanche photodiode (APD) detectors. PL saturation measurements confirmed that all time-resolved PL and $g^{(2)}(\tau)$ measurements of SPEs are taken well below saturation.

For the TEM analysis, the as-grown MoSe₂ QDs@WSe₂ heterostructure films are transferred to Quantifoil Cu TEM grids using a PMMA-assisted wet transfer method[50]. ADF-STEM imaging and STEM-EDS mapping are performed using a dual-corrected Thermo Fisher Titan³ G2 microscope operated at 80 kV. A semi-convergence angle of 30 mrad and a screen current of ~ 50-60 pA are used during the imaging.

### Computational methods

First-principles calculations are performed by using the VASP code[51] with the r²SCAN[52] metaGGA functional and a plane-wave implementation. Parallel GPU computations are used to accelerate the calculations of these large-scale 2D materials systems. The MoSe₂ quantum dots of various sizes are embedded in an 18 × 18 supercell of WSe₂, and calculations are performed using the Γ point in the Brillouin zone. The calculated band gaps for pristine MoSe₂ and WSe₂ are 1.53 and 1.63 eV, respectively. A type-II heterostructure of MoSe₂@WSe₂ is formed. The CBM and the VBM of WSe₂ are higher than those in MoSe₂ by 0.35 and 0.25 eV, respectively. A vacuum thickness of 15 Å is set to prevent the interactions between periodic images. The plane-wave cutoff energy is set to 315 eV. The structural relaxations are performed for all systems

until the force acting on each ion is less than or equal to 0.02 eV/Å. The convergence criteria for total energies in structural relaxations and self-consistent calculations are $10^{-4}$ and $10^{-5}$ eV, respectively.

## Data availability

All data are available in the paper and Supplementary Information. Growth and standard characterization data associated with the samples used in this study is available via ScholarSphere, which is open access at https://doi.org/10.26207/0ddt-qt82. This includes substrate preparation and recipe data for samples grown by MOCVD in the 2DCC-MIP facility and standard characterization data, including AFM images, room temperature Raman/PL spectra and SEM images on the samples.

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

## Acknowledgements

D.J. and G.K. acknowledge primary support for this work by the Air Force Office of Scientific Research (AFOSR) FA2386-20-1-4074 and partial support from FA2386-21-1-4063. The MOCVD samples were grown by B.H., M.L., and J.M.R. in the 2D Crystal Consortium Materials Innovation Platform (2DCC-MIP) facility at Penn State, which is supported by the National Science Foundation under cooperative agreement DMR-2039351. A portion of the sample fabrication, assembly, and characterization were carried out at the Singh Center for Nanotechnology at the University of Pennsylvania, which is supported by the National Science Foundation (NSF) National Nanotechnology Coordinated Infrastructure Program grant NNCI-1542153. The research performed by C.E.S. at the Air Force Research Laboratory was supported by contract award FA807518D0015. J.R.H. acknowledges support from the Air Force Office of Scientific Research (Program Manager Dr. Gernot Pomrenke) under award number FA9550-20RYCOR059. K.J. was supported by a Vagelos Institute of Energy Science and Technology graduate fellowship. J-Y.T. and Q.Y. acknowledge support from the U.S. Department of Energy, Office of Science, under award number DE-SC0023664. This work used resources of the National Energy Research Scientific Computing Center (NERSC), a U.S. Department of Energy Office of Science User Facility located at Lawrence Berkeley National Laboratory, operated under Contract No. DE-AC02-05CH11231 using NERSC award BES-ERCAP0029544. The work of S.B. and N.A. was supported by the NSF CAREER DMR-1654107 grant. H.S.S. acknowledges support from the Institute for Basic Science (IBS-R036-D1). K.Y.M. acknowledges support from the National Research Foundation (NRF-2022R1C1C2009666). N.R.G. gratefully acknowledges support from AFOSR GHz-THz program grant number FA9550-24RYCOR011.

## Author contributions

G.K. and D.J. conceived the measurements and sample fabrication ideas/concepts. J.M.R., B.H., and M.L. conceived and performed the MOCVD-growth of the in-plane $MoSe_2$ QDs@ $WSe_2$ heterostructures and reference ($WSe_2$, $MoSe_2$) monolayers. G.K., S.S., and M.R. performed sample preparation and far-field optical characterizations (temperature dependence, gate-tunable PL, and reflectance measurement down to 80 K). K.J. performed tip-enhanced Raman and KPFM measurements. Q.Y. and J-Y.T. performed DFT-based first-principles calculations. S.B. and N.A performed scanning transmission electron microscopy characterization. C.E.S., under the supervision of J.R.H., performed cryogenic PL (down to 1.6 K), time-resolved PL, and $g^{(2)}(\tau)$ measurements. K.Y.M. and H.S.S. provided the h-BN films CVD-grown on the sapphire substrates. N.R.G. provided valuable comments and feedback on the preparation of the manuscript. G.K. and D.J. wrote the paper with inputs from all co-authors.

## Competing interests

The authors declare competing interests.
