## [Transparent Peer Review file · Nature Communications]

Confinement of Excited States in Two-Dimensional, In-Plane, Quantum Heterostructures

Corresponding Author: Professor Deep Jariwala

Version 0:

Reviewer comments:

Reviewer #1

(Remarks to the Author)

The manuscript by Gwangwoo Kim et al. demonstrated the lateral confinement of excitons in the MoSe₂ QDs embedded within a WSe₂ matrix using sequential epitaxial growth. Various optical spectroscopy techniques reveal size-dependent exciton confinement in MoSe₂ monolayer QDs, resulting in an exciton blue shift (12-40 meV) at lower temperatures compared to continuous MoSe₂ monolayers. Notably, authors also observe single-photon emission ($g^2(0) \sim 0.4$) from the smallest dots at 1.6 K. Overall, this manuscript is well organized and written. Although, the data presented in the manuscript looks promising, however, conclusion is drawn mostly based on the previous studies. The novelty appears somewhat limited. The authors' findings could potentially merit publication in Nature Communications, given that they elucidate the novelty of their results and address a few specific concerns:

1. During cooling, the authors exposed the heterostructure further to the chalcogen source to heal any vacancies. However, have the authors encountered any defects related to adatoms? If not, an explanation is needed.
2. The CPD plot (Fig. 2d) shows potential differences between WSe₂ and MoSe₂. However, this doesn't confirm the type II band alignment unless exact band edge information is provided. Authors need to be careful about their statement. Furthermore, it was expected to have a sharp potential drop for a perfect heterostructure.
3. For imaging the quantum heterostructure, I strongly recommend performing scanning tunneling spectroscopy, which provides more detailed information about the interface and defect states. Furthermore, this can also yield accurate quasiparticle band alignment and band renormalization through lateral heterostructures.
4. In Figures 2e-g, the Raman map also shows A_{1g} Raman mode intensity of MoSe₂ QDs where they aren't expected. The lateral scale in Fig. 2e-g is missing.
5. Why does the WSe₂ photoluminescence peak diminish in Figure 4a as the temperature decreases?
6. In Supplementary Fig. S11b, what is the origin of the lower-energy peak (around 1.62 eV)? Is this due to a trion? If so, why doesn't WSe₂ exhibit a trion peak?
7. In Fig. 4a, what type of defect states are expected? Please explain the origin of these defects otherwise this is in contradiction to the authors' earlier statement about removing chalcogen vacancies through further exposure to the chalcogen source.
8. To demonstrate the confinement effect, is a heterostructure necessary? Because charge density migration may occur due to only dielectric confinement. Please explain in detail. The authors should present the photoluminescence spectra solely for the MoSe₂ QDs and compare them with both pristine MoSe₂ and the heterostructure.

Reviewer #2

(Remarks to the Author)

Kim and coworkers present spectroscopic evidences of lateral excitonic confinement in MoSe₂ quantum dots embedded in WSe₂ matrix. As it is well explained in the abstract, the interest given to TMDs MLs based quantum source is very high as they are very promising materials for a variety of optoelectronic and spintronic device applications due to their optoelectronic properties.

The presented manuscript presents two particularly significant findings: the primary observation is the size dependent blue shift of the optical gap measure in both PL and Reflectance. The authors use a DFT calculation and show that the experimentally observed shifts are of the right order of magnitude. The second significant finding is that at low temperature the smallest MoSe₂ QDs present narrow emission lines, they further demonstrate that the emission acts as a quantum light source.

There are several significant issues that require attention prior to publication, detailed below, and I recommend the

manuscript be reconsidered following revisions.

Major revisions:

- The title of the paper: 'Exciton confinement in two-dimensional, In-plane, Quantum heterostructure' as well as the introduction clearly suggest that the main focus of the paper is the confinement of exciton and the resulting modification of the excitonic energy structure. However, because of the way the experimental data is interpreted, mainly the fact that: (i) DFT calculation is used, and (ii) the actual physical process of exciton confinement is not clearly treated (Why excitons do not diffuse from MoSe₂ toward WSe₂, what happen to exciton excited in WSe₂ and moving into the MoSe₂?) it look to me that the authors are analyzing the data in the light of electronic confinement and how the excitonic features would be modified in a such a quantum dot. Exciton are neutral (bosonic) quasi particle with different dispersion and typical dimensions than electrons/and holes resulting in different behavior when confined. (Does the linear extrapolation presented in 3 e is valid? Does both approaches are equivalent and How? I would like the authors to discuss more deeply theses matters in the paper.

- in Figure 3 b, the 3 red dot are labelled "Experimental data", but as it is described in the caption, they are not experimental data but predicted energy shift using linear extrapolation. This could be quite confusing for the reader and could lead to misinterpretation of the results. This bring to another problem of the manuscript: there is no direct comparisons between the predicted energy shift and the experimentally measured one. Doing so, with either using the PL and/or reflectance found shift using would clarify the accuracy and possible inaccuracies of the model.

- DFT calculation are performed at zero Kelvin, nevertheless all experimental data (PL and Reflectance) that could easily be compared (as 1,6K data present very sharp feature they cannot be easily compare to the theory) are measured at 80K. It is clearly show in the paper itself the energy of the measure optical gap is strongly dependent on the temperature. Can the authors discuss the expected deviation due to this temperature mismatch?

- Due to the laser spot size being much bigger than the Mose2 dot size, The PL data presented in Figure 4 display the signature of both Mose2 dot and the surrounding WSe2. As localized excitonic mode are known to support single photon sources It is then important to be able to guaranty the origin of the measured photons. The authors don't ignore this problem and treats this question, unfortunately two points are bothering me:

o At 1.6K as it can be seen in Figure S11 (measured on reference sample) there is a strong contribution of the defect state of WSe₂ in the 1.65eV -1.7eV frequency range. The authors claim that the WSe₂ defect states appear at 'slightly lower energy position' than the MoSe₂ QDs states the but to me it looks like exactly matching the MoSe₂ QDs mode as described in Figure 4 b (the high intensity peak appearing at 1.66eV). Maybe plotting both curves on the same graph could help.

o The second argument is based on the broad band aspect of the defect related PL observed in the WSe₂ reference sample, but as it is link to defects the aspect of the PL can drastically change from sample to samples and depend on the defect types and density. Furthermore, the MoSe₂/WSe₂ samples are also bound to present more defects than the reference sample, even if only due the presence of the MoSe₂/WSe₂ edges. I also noticed that the 3 PL intensity curves (1.6K) presented in Figure 4 a, b and inset of c are not identical and present different configuration on peaks, such a random configuration suggest to me that defects could be involved. Does for example larger MoSe₂ QDs present (at 1.6 K) such narrow emission lines or not?

I do not suggest that the quantum light source is not originating from the QD, which is possible (and would be a very nice finding), but with the argumentation as it is now, I would suggest a more mitigated claim.

Minor revisions:

- I am slightly questioning the relevance of presenting the PL temperature dependance in the main text as they do not really contribute to the demonstration of the exciton confinement. Maybe focusing on the low temperature data (80K and 1.6K), would make that paper tighter and more impactful.

- Page 13 the optical gap is shifted by 0.11eV maybe a citation could be added to the text to support this.

- At Page 13 the authors claim that the PL intensity ratio of MoSe₂ to WSe₂ is plotted in Supplementary Figure S5b. There is no such figure.

Reviewer #3

(Remarks to the Author)

The growth of monolayer quantum dots (ML-QDs) confined or passivated by forming lateral epitaxy heterostructures is very promising. This kind of ML-QDs have lower density of edge states than the ones without edge passivation. Furthermore, the carrier density in the ML-QDs could be controlled electrically thanks to the formation of the lateral heterostructure. The advantage of this kind of ML-QDs as a photon emitter is obvious. The most important is the formation of the lateral epitaxy heterostructure be confirmed unambiguously.

My concerns and suggestions are as the following,

1, The sequential growth illustrated in Fig 1a is not impossible. Ref-27 of the manuscript reported the growth of MoSe₂/WSe₂ heterojunction with a typical size of several micrometers. This size allows easier and clearer characterization of the consisted MoSe₂ and WSe₂. Here I will appreciate it if the authors can provide further evidence on the formation of the lateral heterostructures with a quantum scale inner MoSe₂.

(i), AFM images of the as-grown samples on the Al₂O₃ substrate, including the samples of MoSe₂ QDs alone grown for different time periods and the samples of MoSe₂ QDs with WSe₂ grown for different time periods. This helps to exclude the vertical van-der-Waals growth of the WSe₂. I have this concern because I noticed that the black-dashed-line circles in Fig 2c correspond to brighter dots in Fig 2b. It seems that the MoSe₂ QDs regions are higher than their surroundings. If this is true, it may indicate that the MoSe₂ QDs are covered by the WSe₂ in this sample.

(ii), TEM images of a second MoSe₂ QDs in the supplementary.

2, About the defect emission of WSe₂. (i) Where is the defect emission from in Fig 4a? Why was this not observed in other samples? (ii) The defect emission in the ref WSe₂ is relatively strong as shown in Fig 11b. Why is this not observed in other samples (Fig 3bc, Fig S6)? (iii) The authors attribute the sharp bands marked in red in Fig 4a as the emission from the 1-min growth MoSe₂ QDs. This band in energy is close to or overlaps with the defect emissions from WSe₂. (Nature Nanotechnology vol 10, pp503–506 (2015)).

3, Fig 4d plots the decay dynamics of the 1.688-eV emission. Can the authors show the decay curve of the 1.6-eV emission?

4, About the trion states in the MoSe₂/WSe₂ heterostructures. The authors concluded that the trions are positive in the heterostructure, as labelled in Fig 3bc and discussed in the main manuscript and the supplementary. According to the type-II band alignment and the electron transfer in the MoSe₂/WSe₂ heterostructures shown in Fig S6, should the trions in Fig 3 be negative (especially for that of the MoSe₂) as no gate voltage was applied.

5, What are the line width of the exciton and trion emissions in the MoSe₂/WSe₂ heterojunction QDs from the curve fitting in Fig 3? Are these numbers larger or smaller than those of the exfoliated samples? Can the authors measure the PL spectrum of the bare MoSe₂ QDs without the following growth of WSe₂? As the edge passivation using lateral epitaxy is one of the most interesting aspects, I suggest the author provide a comparison study of the QDs with and without the WSe₂ passivation.

Author Rebuttal letter:

Point by Point Replies to Reviewer's Comments

Reviewer #1

The manuscript by Gwangwoo Kim et al. demonstrated the lateral confinement of excitons in the MoSe₂ QDs embedded within a WSe₂ matrix using sequential epitaxial growth. Various optical spectroscopy techniques reveal size-dependent exciton confinement in MoSe₂ monolayer QDs, resulting in an exciton blue shift (12-40 meV) at lower temperatures compared to continuous MoSe₂ monolayers. Notably, authors also observe single-photon emission ($g^{(0)} \sim 0.4$) from the smallest dots at 1.6 K. Overall, this manuscript is well organized and written. Although, the data presented in the manuscript looks promising, however, conclusion is drawn mostly based on the previous studies. The novelty appears somewhat limited. The authors' findings could potentially merit publication in Nature Communications, given that they elucidate the novelty of their results and address a few specific concerns:

Reply: We appreciate Reviewer #1's positive comments.

1. During cooling, the authors exposed the heterostructure further to the chalcogen source to heal any vacancies. However, have the authors encountered any defects related to adatoms? If not, an explanation is needed.

Reply: We appreciate this comment and the opportunity to clarify. If chalcogen gas is not supplied during the cooling process, we observed clusters forming on the WSe₂ surface. Our interpretation is that these clusters stem from excess W adatoms that have not yet had the opportunity to diffuse and incorporate into the two-dimensional lattice. Consequently, the cooling in the chalcogen source proves beneficial in reducing the presence of excess metal adatoms on the WSe₂ surface.

We added the following sentence on page 6.

and also to minimize the formation of clusters composed of excess W atoms on the 2D surfaces.

We have added a relevant reference [ACS Nano 9, 2080 (2015)] in the revised manuscript.

23. Eichfeld SM, Hossain L, Lin Y-C, Piasecki AF, Kupp B, Birdwell AG, et al. Highly Scalable, Atomically Thin WSe₂ Grown via Metal-Organic Chemical Vapor Deposition. ACS Nano 2015, 9(2): 2080-2087.

1

2. The CPD plot (Fig. 2d) shows potential differences between WSe₂ and MoSe₂. However, this doesn't confirm the type II band alignment unless exact band edge information is provided. Authors need to be careful about their statement. Furthermore, it was expected to have a sharp potential drop for a perfect heterostructure.

Reply: We agree with this comment. The known spatial resolution of Kelvin Probe Force Microscopy (KPFM) measurement is approximately ~10 nm, but achieving this resolution

proves challenging due to experimental conditions, specific instrumentation, and the external environment. We note that the Au tip diameter used for KPFM measurements is ~20 nm. Because of the analysis limitations, the expected sharp potential drop is not observed, although the TEM image (Figure S2) shows our success in creating a sample with an exceptionally clean interface.

Figure S2. (a) ADF-STEM image of the heterostructures with the 10 min-MoSe₂ QDs. (b, c) Atomic-resolution ADF-STEM image showing an interface between the MoSe₂ QDs and the WSe₂ matrix.

[Image redacted]

We added these changes to the text. (Page 9)

Despite our efforts to establish a clean physical interface (Figure 1c and Supplementary Figure S2), the expected sharp potential drop is not observed due to the limitations of our scanning probe system and Au tip diameter of ~20 nm.

Additionally, we modified the expression related to type II band alignment. (Page 9)

The built-in potential of the quantum heterostructure will drive the electron diffusion from the WSe₂ matrix to the MoSe₂ QDs along the direction of the built-in field, showing its characteristic as typical two semiconductors-based heterostructures.

2

3. For imaging the quantum heterostructure, I strongly recommend performing scanning tunneling spectroscopy, which provides more detailed information about the interface and defect states. Furthermore, this can also yield accurate quasiparticle band alignment and band renormalization through lateral heterostructures.

Reply: We used sapphire substrates to grow quantum heterostructures with the aim of preparing an ensemble of dots embedded in a large monolayer. However, for STM/STS analysis, it becomes necessary to prepare heterostructures on conductive materials like graphene or metal substrates. Although attempts were made to grow the structure on a different substrate, challenges were encountered in optimizing growth conditions. Additionally, achieving a clean surface and interface without polymer and organic residues during the transfer of heterostructures onto another substrate is highly challenging. Therefore, while we agree with the reviewer that this would be a worthwhile experiment, it is beyond the scope of our capabilities and, therefore, presently beyond the scope of this initial work. However, we provide new TEM images, as shown above, that clearly illustrate the sharp heterostructure in small lateral dimensions.

4. In Figures 2e-g, the Raman map also show A_{1g} Raman mode intensity of MoSe₂ QDs where they aren't expected. The lateral scale in Fig. 2e-g is missing.

Reply: The A_{1g} modes of MoSe₂ QDs monolayers are positioned at 240 cm⁻¹, aligning well with previous findings on a CVD-grown MoSe₂ monolayer film [Nanoscale 6, 8949 (2014)]. We have added the relevant references in the revised manuscript.

30. Xia J., Huang X., Liu L. -Z., Wang M., Wang L., Huang B., et al. CVD synthesis of large-area, highly crystalline MoSe₂ atomic layers on diverse substrates and application to photodetectors. *Nanoscale* 2014, 6(15): 8949-8955.

We have provided information about the step size (25 nm) in the caption of Figure 2e. Also, a lateral scale bar has now been added to Figure 2e-g for easier understanding.

3

[Image redacted]

Figure 2. Nanoscale optical and electrical imaging of the MoSe₂ QDs embedded in the WSe₂ matrix. (a) Schematic representation of nanoscale scanning probe techniques (top) on the 5min-MoSe₂ QDs@WSe₂ heterostructures buried in an Au template (bottom: cross-sectional view). The samples are prepared by Au-assisted transfer process^{24, 25}. (b, c) AFM height image and surface potential map of the heterostructures consisting of the 50 nm-sized MoSe₂ QDs and the WSe₂ matrix. (Inset) Height profile marked with white line in Figure b showing no discernible height difference across the crystal proving truly in-plane epitaxy (embedding) of the MoSe₂ QD in the WSe₂ matrix. The MoSe₂ QDs are marked with black dotted circles. (d) Surface potential profile following the green dotted line in Figure c. (e-g) TERS spatial maps of the quantum heterostructures following the red dotted squares in Figures b and c. The TERS images were created within the spectral ranges of 245-255 cm⁻¹ (e, WSe₂ A₁ mode) and 235-245 cm⁻¹ (g, MoSe₂ A_{1g} mode) with a step size of 25 nm. The MoSe₂ QDs

are marked with yellow dotted circles. (f) Overlaid image of Figures (e) and (g). (h) TERS spectra of the MoSe₂ QD regions (red) and the surrounding WSe₂ matrix (blue) as highlighted in the TERS map.

5. Why does the WSe₂ photoluminescence peak diminish in Figure 4a as the temperature decreases?

Reply: As temperature decreases, there is a tendency for increased energy transfer between excited states of the matrix and dot. As shown in Figure 4a, a noticeable reduction in the PL peak of the WSe₂ matrix is observed, indicating the migration of excitons from the WSe₂ matrix to a lower energy defect state or one of the levels of the MoSe₂ quantum dots.

4

6. In Supplementary Fig. S11b, what is the origin of the lower-energy peak (around 1.62 eV)? Is this due to a trion? If so, why doesn't WSe₂ exhibit a trion peak?

Reply: The PL peaks around 1.66 eV and 1.74 eV, depicted in Figure S13b (blue spectrum), are attributed to the defect state and trion, respectively. Figure S12 illustrates that, with decreasing temperature, the trion peak becomes more prominent compared to the neutral exciton. To enhance clarity in peak distinction, a detailed description of each peak has been included in Figure S13.

Figure S13. Temperature-dependent PL spectra of reference MoSe₂ and WSe₂ monolayers. (a) PL spectra of the WSe₂ (top, blue) and MoSe₂ monolayers (bottom, red) measured at 80 K (solid) and room temperature (blurry dot) using a 633 nm-laser excitation in a Linkam stage with a liquid nitrogen supply. (b) Cryogenic PL spectra of WSe₂ (top, blue) and MoSe₂ monolayers (bottom, red) measured at 1.6 K using a 640 nm-laser excitation in a cryostat. For comparison, the cryogenic PL spectrum (middle, black) on the 1-min MoSe₂ QDs@WSe₂ heterostructures was added, as shown in Figure 4b.

5

7. In Fig. 4a, what type of defect states are expected? Please explain the origin of these defects otherwise this is in contradiction to the authors' earlier statement about removing chalcogen vacancies through further exposure to the chalcogen source.

Reply: We performed a chalcogen supply process during the post-growth cooling to minimize defects and vacancies, but healing them completely may be challenging. Additionally, the formation of defects or residues from the transfer process cannot be dismissed. Because of this, in the case of a large area WSe₂ monolayer without MoSe₂ QDs (shown in Figure S13b, blue spectrum), a defect peak is noticeable around 1.66 eV. While the specific atomic structure remains indeterminate, there is a possibility of Mo-based WSe₂ defects doped by residual Mo source used during the MoSe₂ growth process, alongside the presence of general WSe₂ defect structures (such as W or Se vacancies).

We added the following sentence on page 18 to account for the above explanation.

âIn fact, although we performed a chalcogen supply during cooling to minimize defects and vacancies, healing them completely is challenging⁴¹. Additionally, the formation of Mo-based WSe₂ defects introduced by residual Mo sources from the MoSe₂ growth process, or 2D TMD defects possibly formed during the transfer process, cannot be disregarded.â

We have added a relevant reference [ACS Nano 12, 965 (2018)] in the revised manuscript.

41. Lin Y-C, Jariwala B, Bersch BM, Xu K, Nie Y, Wang B, et al. Realizing Large-Scale, Electronic-Grade Two-Dimensional Semiconductors. ACS Nano 2018, 12(2): 965-975.

8. To demonstrate the confinement effect, is a heterostructure necessary? Because charge density migration may occur due to only dielectric confinement. Please explain in detail. The authors should present the photoluminescence spectra solely for the MoSe₂ QDs and compare them with both pristine MoSe₂ and the heterostructure.

Reply: Thanks for the suggestion. This is indeed a valuable suggestion and one that we have attempted to answer comprehensively within our available experimental capabilities. We performed low-temperature PL measurements at 80 K (in a Linkam stage with a liquid nitrogen supply) and 1.6 K (in a cryostat) on samples on MoSe₂ QDs (1, 5, 10 min growth time) shown in Figure S14 (AFM images). As shown in Figure S15a, we observed that the 1-min MoSe₂ QDs (red) exhibited no signal, which is the identical result as on the heterostructures with 1-min MoSe₂ QDs (Figure S12). In contrast, the 5- and 10-min samples (blue and black, respectively) showed discernible peaks corresponding to MoSe₂, indicating successful QD formation. Notably, unlike previous observations in MoSe₂ QDs@WSe₂ heterostructures where no defect or oxidation signals were detected, only showing neutral exciton and trion

peaks (Figure 2b), the current pristine MoSe₂ QD showed clear defect peaks at 80 K for both the 5-min and 10-min samples. These defect states, identified at 800 nm, suggest a potential oxidation issue due to the exposure and lack of passivation of the MoSe₂ QD edges with the WSe₂ matrix. Moreover, an intensity comparison of the defect state intensities relative to those

6

of the neutral exciton and trion shows that the 5-min QDs sample exhibited a more pronounced oxidation peak. This observation could potentially indicate that the 5-min QDs, having a greater number of exposed edges due to their smaller size, are more susceptible to edge oxidation compared to the 10-min QDs. This hypothesis is supported by the assumption that a smaller QD size results in a higher edge-to-basal plane ratio, thereby increasing the likelihood of edge oxidation and preventing any defect emission signals from the sample.

We conducted further cryogenic PL experiments on the three kinds of samples within a cryostat at 1.6 K temperature (Figure S15b). For the 5- and 10-min samples, a defect states emission was dominantly observed, with no detectable signal from neutral excitons. Conversely, the 1-min QDs sample exhibited distinct quantum emission signals at energies of ~1.7 eV. This emission, which was observed even without incorporating a WSe₂ matrix, highlights the influence of quantum-confined excitons within the MoSe₂ QDs. This observation also highlights that extreme spatial confinement is critical as it not only helps induce stronger quantum confinement but also allows the separation between energy levels to be large enough that unpassivated or oxidized edge levels do not necessarily prevent quantum emission. More simply put, this observation indicates that QDs with smaller sizes are less impacted by edge defect states, which otherwise induce classical emission in larger QDs. Despite this observation, the emission intensity didn't reach the levels seen in heterostructures, further suggesting that despite preservation of quantum confinement at small sizes, unpassivated edges do lead to much lower emission intensities, indicating enhanced non-radiative recombination at the edges. A detailed understanding of the effect of size vs edge effects on quantum emission from as-grown MoSe₂ QDs without a matrix will require more detailed growth and measurements and will be the subject of our future work.

Next, we tried to measure the second-order coherence function(g_2) for the sharp emissions, but we couldn't get any reliable results because the signal was too weak, once again suggesting the effect of lack of edge passivation/edge oxidation. This difficulty indicates how important it is to passivate the edges of the MoSe₂ QDs with the WSe₂ matrix in the planar heterostructures.

Figure S14. AFM height images of the MoSe₂ QDs with different growth times (a: 1 min, b: 5 min, c: 10 min) without the WSe₂ matrix, which were measured on sapphire substrates.

7

Figure S15. Low temperature PL spectra of MoSe₂ QDs without WSe₂ matrix. (a) PL spectra of the MoSe₂ QDs with different growth time of 10 mins (top, black), 5 mins (middle, blue), and 1 min (bottom, red) measured at 80 K using a 633 nm-laser excitation in a Linkam stage with a liquid nitrogen supply. (b) Cryogenic PL spectra of the MoSe₂ QDs with different growth time of 10 mins (top, black), 5 mins (middle, blue), and 1 min (bottom, red) measured at 1.6 K using a 640 nm-laser excitation in a cryostat.

We added the comparative results on the MoSe₂ QD into the manuscript (Page 18) and Supplementary Information

Furthermore, to clarify the quantum emission from the MoSe₂ QDs, we prepared control samples comprising only MoSe₂ QDs with different growth times without the WSe₂ matrix (Supplementary Figure S14) and performed cryogenic PL measurement on the samples at 1.6 K (Supplementary Figure S15). The 1-min MoSe₂ QDs (red line) showed sharp emission at ~1.7 eV, indicative of quantum-confined excitons. However, only broad defect state emissions were observed in the 5- and 10-min samples (blue and black lines). This observation indicates that QDs with smaller sizes are less impacted by edge defect states, which otherwise induce classical emission in larger QDs. Note that the intensity of quantum emission didn't reach the levels seen in MoSe₂ QD/WSe₂ heterostructures. This observation emphasizes the importance of edge passivation. Our findings illustrate that in quantum heterostructures, light absorption in the WSe₂ matrix facilitates exciton confinement within the MoSe₂ QDs, enhancing light emission and providing insights into the interaction between material interfaces and quantum confinement. This also suggests and merits future work on a detailed understanding of the effect of QD size and edge chemistry on quantum emission from as-grown MoSe₂ QDs.

Reviewer #2

Kim and coworkers present spectroscopic evidences of lateral excitonic confinement in MoSe₂ quantum dots embedded in WSe₂ matrix. As it is well explained in the abstract, the interest given to TMDs MLs based quantum source is very high as they are very promising materials for a variety of optoelectronic and spintronic device applications due to their optoelectronic properties.

The presented manuscript presents two particularly significant findings: the primary observation is the size dependent blue shift of the optical gap measure in both PL and Reflectance. The authors use a DFT calculation and show that the experimentally observed shifts are of the right order of magnitude. The second significant finding is that at low temperature the smallest MoSe₂ QDs present narrow emission lines, they further demonstrate that the emission act as a quantum light source.

There are several significant issues that require attention prior to publication, detailed below, and I recommend the manuscript be reconsidered following revisions.

Reply: We would like to thank Reviewer #2 for the positive comments.

Major revisions:

1. The title of the paper: "Exciton confinement in two-dimensional, In-plane, Quantum heterostructure" as well as the introduction clearly suggest that the main focus of the paper is the confinement of exciton and the resulting modification of the excitonic energy structure. However, because of the way the experimental data is interpreted, mainly the fact that: (i) DFT calculation is used, and (ii) the actual physical process of exciton confinement is not clearly treated (Why excitons do not diffuse from MoSe₂ toward WSe₂, what happen to exciton excited in WSe₂ and moving into the MoSe₂?) it look to me that the authors are analyzing the data in the light of electronic confinement and how the excitonic features would be modified in a such a quantum dot. Exciton are neutral (bosonic) quasi particle with different dispersion and typical dimensions than electrons/and holes resulting in different behavior when confined. (Does the linear extrapolation present in 3e is valid? Does both approaches are equivalent and how? I would like the authors to discuss more deeply these matters in the paper.

Reply: We agree with the reviewer's comment that electronic confinement and excitonic confinement are not completely equivalent. Based on the Bethe-Salpeter Equation, the first-order effect of quantum dot size on excitonic energies is the shift of energies of electron and hole states. Furthermore, the quantum well size may affect the excitonic interactions among these states and thus induce further shift or broadening of excitonic peaks.

As discussed in the manuscript, to approximate the excitonic effects and the underestimation of band gaps by DFT, we increase the calculated band energy differences by 0.11 eV (which is the energy difference between the calculated band gap and the measured optical gap of MoSe₂) to evaluate the optical transition energies roughly. Although our DFT calculations do not

9

explicitly include the excitonic effects, which can be significant in two-dimensional materials, the consistent results suggest that the change in the band energies of those electronic states involved in optical transitions (electronic confinement) may be dominant in the observed shift of transition energies.

Note that ab initio computations of excitonic effects (based on the Bethe-Salpeter Equation formalism) for quantum dots with exceedingly large sizes (>1000 atoms) are beyond our computational capability and, therefore, are out of the scope of our current work.

We added the following discussions on page 15.

"Note that the effects of quantum dot confinement on electrons and excitons are not completely equivalent. Based on the Bethe-Salpeter Equation, the first-order effect of the quantum dot size on excitonic energies involves the shifting of electron and hole state energies. Furthermore, the size of the quantum well may affect the excitonic interactions among these states and thus induce further shifts or broadening of excitonic peaks."

In addition, we have also changed the title of the manuscript to address reviewer's comment:

New title: "Confinement of Excited States in Two-Dimensional, In-plane, Quantum heterostructures"

2. In Figure 3e, the 3 red dot are labelled "Experimental data", but as it is described in the caption, they are not experimental data but predicted energy shift using linear extrapolation. This could be quite confusing for the reader and could lead to misinterpretation of the results. This bring to another problem of the manuscript: there is no direct comparisons between the predicted energy shift and the experimentally measured one. Doing so, with either using the PL and/or reflectance found shift using would clarify the accuracy and possible inaccuracies of the model.

Reply: The red dots shown in Figure 3e (instead of 3d as noted by the Reviewer) are indeed the predicted energy shift using linear extrapolation for the 15, 30, and 50-nm MoSe₂ QDs. To avoid confusion and further clarify this point, we revised Figure 3e accordingly.

As mentioned on Page 13, we calculated the energy shifts of the 10, 30, and 50-nm MoSe₂ QDs based on the extrapolation. The predicted energy shifts of 30 and 50-nm dots are 20 and

12 meV, respectively, and the experimental data of energy shifts for 5- and 10-min MoSe2 QDs are 20 and 15 meV, respectively. This provides a direct comparison between the predictions and experimental data.

10

[Image redacted]

Figure 3d. Relation between the energy shifts on size of the MoSe2 QDs embedded in the WSe2 matrix. The dashed blue line represents the linear fit of the energy shifts with respect to the inverse size of QDs. The blue dots represent the computationally estimated energy shifts in the QDs with sizes ranging from 2.7 to 5.3 nm. The three red dots are predicted energy shifts in 15, 30, and 50-nm quantum dots using linear extrapolation, are well matched quantitatively with experimental data in c. The experimentally determined optical band gap of pristine MoSe2 is set as zero of energy and associated with an infinitely large ($R = \hat{a}$) dot.

3. DFT calculation are performed at zero Kelvin, nevertheless all experimental data (PL and Reflectance) that could easily be compared (as 1.6K data present very sharp feature they cannot be easily compare to the theory) are measured at 80K. It is clearly show in the paper itself the energy of the measure optical gap is strongly dependent on the temperature. Can the authors discuss the expected deviation due to this temperature mismatch?

Reply: We agree with the referee that DFT calculations are performed without including any temperature effect. To computationally study the temperature-dependent excitonic effects, electron-phonon interaction has to be explicitly included (for instance, Phys. Rev. B 93, 155435 (2016)). This is not computationally accessible for large-scale QD systems and, therefore, is out of the scope of our current work.

We added the following discussions in the page 15.

Additionally, it is important to note that the exciton-phonon interactions may influence both the decrease in exciton peak energy and the line width, while our computational results do not consider the effect of temperature. Nevertheless, it is expected that temperature effects play a minimal role in the relative shift of the excitonic peaks concerning quantum dot size, which is the primary focus of this computational study.

4. Due to the laser spot size being much bigger than the MoSe2 dot size, The PL data presented in Figure 4 display the signature of both MoSe2 dot and the surrounding WSe2. As localized excitonic mode are known to support single photon sources It is then important to be able to

11

guaranty the origin of the measured photons. The authors don't ignore this problem and treats this question, unfortunately two points are bothering me:

o At 1.6K as it can be seen in Figure S11 (measured on reference sample) there is a strong contribution of the defect state of WSe2 in the 1.65eV -1.7eV frequency range. The authors claim that the WSe2 defect states appear at a slightly lower energy position than the MoSe2 QDs states the but to me it looks like exactly matching the MoSe2 QDs mode as described in Figure 4 b (the high intensity peak appearing at 1.66eV). Maybe plotting both curves on the same graph could help.

Reply: We are grateful to the reviewer for this suggestion. For clear comparisons, we have now added the cryogenic PL spectrum measured on the 1-min MoSe2 QDs@WSe2 heterostructures (Figure 4b). We also replaced the Supplementary Figure S13 with the following modified figure:

[Image redacted]

Figure S13. Temperature-dependent PL spectra of reference MoSe2 and WSe2 monolayers. (a) PL spectra of the WSe2 (top, blue) and MoSe2 monolayers (bottom, red) measured at 80 K (solid) and room temperature (blurry dot) using a 633 nm-laser excitation in a Linkam stage with a liquid nitrogen supply. (b) Cryogenic PL spectra of WSe2 (top, blue) and MoSe2 monolayers (bottom, red) measured at 1.6 K using a 640 nm-laser excitation in a cryostat. For comparison, the cryogenic PL spectrum (middle, black) on the 1-min MoSe2 QDs@WSe2 heterostructures was added, shown in Figure 4b.

12

5. The second argument is based on the broad band aspect of the defect related PL observed in the WSe2 reference sample, but as it is link to defects the aspect of the PL can drastically change from sample to samples and depend on the defect types and density. Furthermore, the MoSe2@WSe2 samples are also bound to present more defects than the reference sample, even

if only due the presence of the MoSe₂@WSe₂ edges. I also noticed that the 3 PL intensity curves (1.6K) presented in Figure 4 a, b and inset of c are not identical and present different configuration on peaks, such a random configuration suggest to me that defects could be involved. Does for example larger MoSe₂ QDs present (at 1.6 K) such narrow emission lines or not?

Reply: Thanks for the suggestion. We understand your concerns about the potential origin of PL emission in defects, particularly at the edges of MoSe₂ QDs@WSe₂. In our study, we specifically addressed the issue of defects, particularly at the interface between MoSe₂ QDs and the WSe₂ matrix. Through TEM analysis (Figure S2), we showed that there are almost no defects and remarkably clean at this interface.

[Image redacted]

Figure S2. (a) ADF-STEM image of the heterostructures with the 10 min-MoSe₂ QDs. (b, c) Atomic-resolution ADF-STEM image showing an interface between the MoSe₂ QDs and the WSe₂ matrix.

Regarding comparison with MoSe₂ QDs, we perform low-temperature PL measurements at 80 K (in a Linkam stage with a liquid nitrogen supply) and 1.6 K (in a cryostat) on samples on MoSe₂ QDs (1, 5, 10 min growth time) shown in Figure S14 (AFM images). As shown in Figure S15a, we observed that the 1-min MoSe₂ QDs (red) exhibited no signal, which is the identical result as on the heterostructures with 1-min MoSe₂ QDs (Figure S12). In contrast, the 5- and 10-min samples (blue and black, respectively) showed discernible peaks corresponding to MoSe₂, indicating successful QD formation. Notably, unlike our observations in MoSe₂ QDs@WSe₂ heterostructures where no defect or oxidation signals were detected, only showing neutral exciton and trion peaks (Figure 2b), the current pristine MoSe₂ QD showed clear defect peaks at 80 K for both the 5-min and 10-min samples. These defect states, identified at 800 nm, suggest a potential oxidation issue likely due to the exposure + oxidation and/or lack of passivation of the MoSe₂ QD edges within the WSe₂ matrix. Moreover, a comparison of the defect state intensities relative to those of the neutral exciton and trion shows that the 5-min

13
QD sample exhibits a more pronounced oxidation peak. This observation could potentially indicate that the 5-min QDs, having a greater number of exposed edges due to their smaller size are therefore more susceptible to edge oxidation compared to the 10-min QDs. This hypothesis is supported by the assumption that a smaller QD size results in a higher edge-to-basal plane ratio, thereby increasing the likelihood of edge oxidation within the same beam size. While this hypothesis is valid for 10 min and 5 min QD samples, the observations for 1 min QD suggest a more complex picture.

We also perform cryogenic PL experiments on the three kinds of samples within a cryostat at 1.6 K temperature (Figure S15b). For the 5- and 10-min samples, defect state emission was dominantly observed, with no detectable signal from neutral excitons. Conversely, the 1-min QDs sample exhibited distinct sharp peaks indicative of quantum emission signals at energies of ~1.7 eV. This emission, which was observed even without incorporating MoSe₂ QDs in a WSe₂ matrix, highlights the degree of quantum-confined excitons within the MoSe₂ QDs for 1 min growth time. This observation indicates that QDs with smaller sizes are less impacted by edge defect states, which otherwise induce classical emission in larger QDs. This observation also highlights that extreme spatial confinement is critical as it not only helps induce stronger quantum confinement but also allows the separation between energy levels to be large enough that unpassivated or oxidized edge levels do not necessarily prevent quantum emission. More simply put, this observation indicates that QDs with smaller sizes are less impacted by edge defect states, which otherwise induce classical emission in larger QDs. Despite this observation, the emission intensity didn't reach the levels seen in heterostructures, further suggesting that despite preservation of quantum confinement at small sizes, unpassivated edges do lead to much lower emission intensities, indicating enhanced non-radiative recombination at the edges. A detailed understanding of the effect of size vs edge effects on quantum emission from as-grown MoSe₂ QDs without a matrix will require more detailed growth and measurements and will be the subject of our future work.

Next, we tried to measure the second-order coherence function(g_2) for the sharp emissions, but we couldn't get any reliable results because the signal was too weak, once again suggesting the effect of lack of edge passivation/edge oxidation. This difficulty indicates how important it is to passivate the edges of the MoSe₂ QDs with the WSe₂ matrix in the planar heterostructures.

14

[Image redacted]

Figure S14. AFM height images of the MoSe₂ QDs with different growth times (a: 1 min,

b: 5 min, c: 10 min) without the WSe₂ matrix, which were measured on sapphire substrates.

[Image redacted]

Figure S15. Low temperature PL spectra of MoSe₂ QDs without WSe₂ matrix. (a) PL spectra of the MoSe₂ QDs with different growth time of 10 mins (top, black), 5 mins (middle, blue), and 1 min (bottom, red) measured at 80 K using a 633 nm-laser excitation in a Linkam stage with a liquid nitrogen supply. (b) Cryogenic PL spectra of the MoSe₂ QDs with different growth time of 10 mins (top, black), 5 mins (middle, blue), and 1 min (bottom, red) measured at 1.6 K using a 640 nm-laser excitation in a cryostat.

We added the comparative results on the MoSe₂ QD into the manuscript (Page 18) and Supplementary Information

Furthermore, to clarify the quantum emission from the MoSe₂ QDs, we prepared control samples comprising only MoSe₂ QDs with different growth times without the WSe₂ matrix (Supplementary Figure S14) and performed cryogenic PL measurement on the samples at 1.6 K (Supplementary Figure S15). The 1-min MoSe₂ QDs (red line) showed sharp emission at ~1.7 eV, indicative of quantum-confined excitons. However, only broad defect state emissions were observed in the 5- and 10-min samples (blue and black lines). This observation indicates that QDs with smaller sizes are less impacted by edge defect states, which otherwise induce classical emission in larger QDs. Note that the intensity of quantum emission didn't reach the levels seen in MoSe₂ QD/WSe₂ heterostructures. This observation emphasizes the importance of edge passivation. Our findings illustrate that in quantum heterostructures, light absorption in the WSe₂ matrix facilitates exciton confinement within the MoSe₂ QDs, enhancing light emission and providing insights into the interaction between material interfaces and quantum confinement. This also suggests and merits future work on a detailed understanding of the effect of QD size and edge chemistry on quantum emission from as-grown MoSe₂ QDs.

15

Minor revisions:

6. I am slightly questioning the relevance of presenting the PL temperature dependence in the main text as they do not really contribute to the demonstration of the exciton confinement. Maybe focusing on the low temperature data (80K and 1.6K), would make that paper tighter and more impactful.

Reply: Thank you for the suggestion. Based on Reviewer #2's suggestion, we have relocated the temperature-dependent PL results to the Supplementary Information (Figure S6) and reformatted them to display the 80K spectra within Figure 2.

[Image redacted]

Figure 3. Exciton confinement of the MoSe₂ QDs embedded in the WSe₂ matrix. (a) Schematic representation of the in-plane MoSe₂ QDs@WSe₂ heterostructures encapsulated in top and bottom h-BN tri-layers. The Mo, W, Se, B, and N atoms are represented in red, green, yellow, pink, and blue, respectively. (b) PL spectra of the heterostructures with different growth times of the MoSe₂ QDs (top: 5min, bottom: 10 min) measured at 80 K temperature. These spectra were obtained by a 633 nm-CW laser with an excitation power of 20 μ W and a 50x lens with 0.35 NA. (c) Comparison of PL energy position of the main neutral excitons of MoSe₂ and WSe₂ in the heterostructure samples (red: 5 min, blue: 10 min) with reference monolayers (black). The points are plotted from the peak positions in Figure b. The reference

16

MoSe₂ and WSe₂ monolayers are prepared in the same MOCVD chamber. (d) Relation between the energy shift and the size of the MoSe₂ QDs embedded in the WSe₂ matrix. The dashed blue line represents the linear fit of the energy shift with respect to the inverse size of QDs. The blue dots represent the computationally estimated energy shift in the QDs with sizes ranging from 2.7 to 5.3 nm. The three red dots are predicted energy shifts in 15, 30, and 50-nm quantum dots using linear extrapolation. The experimentally determined optical band gap of pristine MoSe₂ is set as zero energy and associated with an infinitely large ($R = \infty$) quantum dot.

[Image redacted]

Figure S6. Temperature-dependent PL spectra (280 K to 80 K) of the heterostructures with

different growth times of the MoSe₂ QDs (a: 5 min, b: 10 min). These spectra were obtained by continuous 633 nm-laser with an excitation power of 20 μ W and a 50x lens with 0.35 NA.

7. Page 13 the optical gap is shifted by 0.11 eV maybe a citation could be added to the text to support this.

Reply: Thank you for the suggestion. We added a citation to support this. (J. Phys. Chem. C 126, 21022 (2022))

36. Jones L. A. H., Xing Z., Swallow J. E. N., Shiel H., Featherstone T. J., Smiles M. J., et al. Band Alignments, Electronic Structure, and Core-Level Spectra of Bulk Molybdenum Dichalcogenides (MoS₂, MoSe₂, and MoTe₂). The Journal of Physical Chemistry C 2022, 126(49): 21022-21033.

17

8. At Page 13 the authors claim that the PL intensity ratio of MoSe₂ to WSe₂ is plotted in Supplementary Figure S5b. There is no such figure.

Reply: Thank you for the correction. The sentence has been revised as follow:

Supplementary Figure S5b to Supplementary Figure S5a in the page 12.

18

Reviewer #3

The growth of monolayer quantum dots (ML-QDs) confined or passivated by forming lateral epitaxy heterostructures is very promising. This kind of ML-QDs have lower density of edge states than the ones without edge passivation. Furthermore, the carrier density in the ML-QDs could be controlled electrically thanks to the formation of the lateral heterostructure. The advantage of this kind of ML-QDs as a photon emitter is obvious. The most important is the formation of the lateral epitaxy heterostructure be confirmed unambiguously.

Reply: We appreciate Reviewer #3's comments and in particular the reviewer's reflection of the promise and importance of our work.

My concerns and suggestions are as the following,

1, The sequential growth illustrated in Fig 1a is not impossible. Ref-27 of the manuscript reported the growth of MoSe₂@WSe₂ heterojunction with a typical size of several micrometers. This size allows easier and clearer characterization of the consisted MoSe₂ and WSe₂. Here I will appreciate it if the authors can provide further evidence on the formation of the lateral heterostructures with a quantum scale inner MoSe₂.

(i) AFM images of the as-grown samples on the Al₂O₃ substrate, including the samples of MoSe₂ QDs alone grown for different time periods and the samples of MoSe₂ QDs with WSe₂ grown for different time periods. This helps to exclude the vertical van-der-walls growth of the WSe₂. I have this concern because I noticed that the black-dashed-line circles in Fig 2c correspond to brighter dots in Fig 2b. It seems that the MoSe₂ QDs regions are higher than their surroundings. If this is true, it may indicate that the MoSe₂ QDs are covered by the WSe₂ in this sample.

Reply: Thanks for the valuable suggestion. We have added some AFM images for a more comprehensive comparison. As shown in Figure S14, it is clearly evident that the MoSe₂ QDs are formed with a noticeable increase in their size correlating with longer growth times.

Conversely, in samples comprising heterostructures (MoSe₂ QDs@WSe₂ matrix, Figure R1), the quantum dots were indistinguishable, and instead, what appeared to be a continuous film was observed. Therefore, to address this challenge and differentiate between MoSe₂ QDs and WSe₂ within the heterostructure, we performed KPFM and near-field analysis on an Au substrate, as shown in Figure 2.

19

[Image redacted]

Figure S16. AFM height images of the MoSe₂ QDs with different growth times (a: 1 min, b: 5 min, c: 10 min) without the WSe₂ matrix, which were measured on sapphire substrates.

[Image redacted]

Figure R1. AFM height images of the MoSe₂ QDs@WSe₂ heterostructures with different QDs growth times (a: 1 min, b: 5 min, c: 10 min), which were measured on sapphire substrates.

(ii), TEM images of a second MoSe₂ QDs in the supplementary.

Reply: We performed the TEM measurement on the holey carbon grids, which are suitable only for supporting continuous or semi-continuous films and not for isolated small entities such as MoSe₂ QDs. If we try to transfer QDs, they will just fall through the holes in the TEM grid support. However, to demonstrate the planar heterostructure between MoSe₂ and WSe₂, we added some TEM images for 10 min-MoSe₂ QDs in Supplementary Figure S2, which successfully depicted this in-plane configuration.

[Image redacted]

20

Figure S2. (a) ADF-STEM image of the heterostructures with the 10 min-MoSe₂ QDs. (b, c) Atomic-resolution ADF-STEM image showing an interface between the MoSe₂ QDs and the WSe₂ matrix.

2, About the defect emission of WSe₂. (i) Where is the defect emission from in Fig 4a? Why was this not observed in other samples? (ii) The defect emission in the ref WSe₂ is relatively strong as shown in Fig 11b. Why is this not observed in other samples (Fig 3bc, Fig S6)? (iii) The authors attribute the sharp bands marked in red in Fig 4a as the emission from the 1-min growth MoSe₂ QDs. This band in energy is close to or overlaps with the defect emissions from WSe₂. (Nature Nanotechnology vol 10, pp503-506 (2015)).

Reply: Regarding the defect emission, we performed a chalcogen supply process during the cooling to minimize defects and vacancies, but healing them completely may be challenging. Additionally, the formation of defects or residues from the transfer process cannot be dismissed. Because of this, in the case of a large area WSe₂ monolayer without MoSe₂ QDs (shown in Figure S11b, blue spectrum), a defect peak is noticeable around 1.66 eV. While the specific atomic structure remains indeterminate, there is a possibility of Mo-based WSe₂ defects doped by residual Mo source used during the MoSe₂ growth process, alongside the presence of general WSe₂ defect structures (such as W or Se vacancies).

We added the following sentence in the page 18.

In fact, although we performed a chalcogen supply during cooling to minimize defects and vacancies, healing them completely is challenging⁴¹. Additionally, the formation of Mo-based WSe₂ defects introduced by residual Mo sources from the MoSe₂ growth process, or 2D TMD defects possibly formed during the transfer process, cannot be disregarded.

We have added a relevant reference [ACS Nano 12, 965 (2018)] in the revised manuscript.

41. Lin Y-C, Jariwala B, Bersch BM, Xu K, Nie Y, Wang B, et al. Realizing Large-Scale, Electronic-Grade Two-Dimensional Semiconductors. ACS Nano 2018, 12(2): 965-975.

The paper (Nat. Nanotechnol. 10, 503 (2015)) shows quantum emission in an energy range of 1.63-1.70 eV. We agree that the emission energy partially overlaps with that of our heterostructure. However, to clarify this point, we prepared samples composed of only MoSe₂ QDs (different growth times) without incorporating the WSe₂ matrix and conducted cryogenic PL measurements on the samples. We observed quantum emission at a comparable energy level. A comprehensive response to this observation can be found in the section addressing comment #5 below.

21

3, Fig 4d plots the decay dynamics of the 1.688-eV emission. Can the authors show the decay curve of the 1.6-eV emission?

Reply: We added the decay curve for the 1.60 eV emission shown in Figure 4d, inset. This measurement was conducted under identical conditions, showing a notably long lifetime of 16.2 ns when compared to the 1.688 eV emission. Therefore, we suggest that this extended lifetime is attributed to defect states.

[Image redacted]

Figure S15. TRPL spectrum for the defect states of the MoSe₂ QDs (1.60 eV), shown in Figure 4d, inset. The time-resolved PL data (blue line) are convoluted (black line) with the

instrument response function, using an exponential function $I = A \cdot \exp(-t/\tau)$. We added the above result in the Supplementary Information.

4, About the trion states in the MoSe₂@WSe₂ heterostructures. The authors concluded that the trions are positive in the heterostructure, as labelled in Fig 3bc and discussed in the main manuscript and the supplementary. According to the type-II band alignment and the electron transfer in the MoSe₂@WSe₂ heterostructures shown in Fig S6, should the trions in Fig 3 be negative (especially for that of the MoSe₂) as no gate voltage was applied.

Reply: Thanks for the comment. In the gate-control PL measurement, we observed a clear change in trion peak intensity when negative bias was applied. Therefore, we determined that positively charged trions were predominant. Based on Reviewer #2's opinion, we revised the terminology for the trion peak to XT^{\pm} and refrained from specifying the charge as either positive or negative.

5, What are the line width of the exciton and trion emissions in the MoSe₂@WSe₂ heterojunction QDs from the curve fitting in Fig 3? Are these numbers larger or smaller than those of the exfoliated samples? Can the authors measure the PL spectrum of the bare MoSe₂ QDs without the following growth of WSe₂? As the edge passivation using lateral epitaxy is

one of the most interesting aspects, I suggest the author provide a comparison study of the QDs with and without the WSe₂ passivation.

Reply: Thanks for the suggestion. The PL spectra presented in Figure 3, obtained at a temperature of 80K, showed that the linewidths for both exciton and trion emissions range between approximately 25 to 30 meV, which closely match those observed in the exfoliated samples.

Regarding comparison with QDs without WSe₂ matrix, we perform low-temperature PL measurements at 80 K (in a Linkam stage with a liquid nitrogen supply) and 1.6 K (in a cryostat) on samples on MoSe₂ QDs (1, 5, 10 min growth time) shown in Figure S14 (AFM images). As shown in Figure S15a, we observed that the 1-min MoSe₂ QDs (red) exhibited no signal, which is the identical result as on the heterostructures with 1-min MoSe₂ QDs (Figure S12). In contrast, the 5- and 10-min samples (blue and black, respectively) showed discernible peaks corresponding to MoSe₂, indicating successful QD formation. Notably, unlike our observations in MoSe₂ QDs@WSe₂ heterostructures where no defect or oxidation signals were detected, only showing neutral exciton and trion peaks (Figure 2b), the current pristine MoSe₂ QD showed clear defect peaks at 80 K for both the 5-min and 10-min samples. These defect states, identified at 800 nm, suggest a potential oxidation issue likely due to the exposure + oxidation and/or lack of passivation of the MoSe₂ QD edges within the WSe₂ matrix. Moreover, a comparison of the defect state intensities relative to those of the neutral exciton and trion, shows that the 5-min QD sample exhibits a more pronounced oxidation peak. This observation could potentially indicate that the 5-min QDs, having a greater number of exposed edges due to their smaller size are therefore more susceptible to edge oxidation compared to the 10-min QDs. This hypothesis is supported by the assumption that a smaller QD size results in a higher edge-to-basal plane ratio, thereby increasing the likelihood of edge oxidation within the same beam size. While this hypothesis is valid for 10 min and 5 min QD samples, the observations for 1 min QD suggest a more complex picture.

We also perform cryogenic PL experiments on the three kinds of samples within a cryostat at 1.6 K temperature (Figure S15b). For the 5- and 10-min samples, defect state emission was dominantly observed, with no detectable signal from neutral excitons. Conversely, the 1-min QDs sample exhibited distinct sharp peaks indicative of quantum emission signals at energies of ~ 1.7 eV. This emission, which was observed even without incorporating MoSe₂ QDs in a WSe₂ matrix, highlights the degree of quantum-confined excitons within the MoSe₂ QDs for 1 min growth time. This observation indicates that QDs with smaller sizes are less impacted by edge defect states, which otherwise induce classical emission in larger QDs. This observation indicates that QDs with smaller sizes are less impacted by edge defect states, which otherwise induce classical emission in larger QDs. This observation also highlights that extreme spatial confinement is critical as it not only helps induce stronger quantum confinement but also allows the separation between energy levels to be large enough that unpassivated or oxidized edge levels do not necessarily prevent quantum emission. More simply put, this observation

23

indicates that QDs with smaller sizes are less impacted by edge defect states, which otherwise induce classical emission in larger QDs. Despite this observation, the emission intensity didn't reach the levels seen in heterostructures, further suggesting that despite preservation of quantum confinement at small sizes, unpassivated edges do lead to much lower emission intensities, indicating enhanced non-radiative recombination at the edges. A detailed understanding of the effect of size vs edge effects on quantum emission from as-grown MoSe₂ QDs without a matrix will require more detailed growth and measurements and will be the

subject of our future work.

Next, we tried to measure the second-order coherence function (g_2) for the sharp emissions, but we couldn't get any reliable results because the signal was too weak, once again suggesting the effect of lack of edge passivation/edge oxidation. This difficulty indicates how important it is to passivate the edges of the MoSe₂ QDs with the WSe₂ matrix in the planar heterostructures.

[Image redacted]

Figure S14. AFM height images of the MoSe₂ QDs with different growth times (a: 1 min, b: 5 min, c: 10 min) without the WSe₂ matrix, which were measured on sapphire substrates.

[Image redacted]

Figure S15. Low temperature PL spectra of MoSe₂ QDs without WSe₂ matrix. (a) PL spectra of the MoSe₂ QDs with different growth time of 10 mins (top, black), 5 mins (middle, blue), and 1 min (bottom, red) measured at 80 K using a 633 nm-laser excitation in a Linkam stage with a liquid nitrogen supply. (b) Cryogenic PL spectra of the MoSe₂ QDs with different

24

growth time of 10 mins (top, black), 5 mins (middle, blue), and 1 min (bottom, red) measured at 1.6 K using a 640 nm-laser excitation in a cryostat.

We added the comparative results on the MoSe₂ QD into the manuscript (Page 18) and Supplementary Information

Furthermore, to clarify the quantum emission from the MoSe₂ QDs, we prepared control samples comprising only MoSe₂ QDs with different growth times without the WSe₂ matrix (Supplementary Figure S14) and performed cryogenic PL measurement on the samples at 1.6 K (Supplementary Figure S15). The 1-min MoSe₂ QDs (red line) showed sharp emission at ~1.7 eV, indicative of quantum-confined excitons. However, only broad defect state emissions were observed in the 5- and 10-min samples (blue and black lines). This observation indicates that QDs with smaller sizes are less impacted by edge defect states, which otherwise induce classical emission in larger QDs. Note that the intensity of quantum emission didn't reach the levels seen in MoSe₂ QD/WSe₂ heterostructures. This observation emphasizes the importance of edge passivation. Our findings illustrate that in quantum heterostructures, light absorption in the WSe₂ matrix facilitates exciton confinement within the MoSe₂ QDs, enhancing light emission and providing insights into the interaction between material interfaces and quantum confinement. This also suggests and merits future work on a detailed understanding of the effect of QD size and edge chemistry on quantum emission from as-grown MoSe₂ QDs.

25

Version 1:

Reviewer comments:

Reviewer #1

(Remarks to the Author)

In the revised manuscript, the authors provide new experimental evidence and more detailed interpretations to address my previous comments. In general, I agree with most of the explanations provided by the authors. Based on the current revisions, I believe the manuscript is suitable for publication after addressing the following issues:

I still have concerns regarding the interpretation of the PL spectra (previous comment 6).

1. It is well known that TMDs exhibit complex excitonic states, such as neutral excitons, trions, biexcitons, and localized states. The authors need to be very careful in assigning these peaks. Specifically, in the blue curve, it is unclear why the WSe₂ neutral exciton (which was not shown) and trion (1.74 eV) peak energies are so different from the other data (Figure 3b, Figure S12), where the neutral exciton and trion positions are approximately 1.74 eV and 1.70 eV, respectively. If the authors claim that at low temperatures the energy is upshifted (even though for MoSe₂ it does not shift much as we can see in Figure 3b and Figure S13b), then they must show the presence of the neutral exciton, which would be approximately 30 meV higher than the trion (Nature Materials 2013, 12, 207-211; ACS Nano 2021, 15, 2849-2857). In contrast, the MoSe₂ peaks are distinctly visible and consistent. Please address this discrepancy with a more thorough explanation to support the peak assignment.

2. Why is each individual layer's FWHM quite broad even at 1.6 K compared to MoSe₂QDs@WSe₂ in Figure S13? Please provide a comprehensive analysis to clarify this observation.

Regarding the use of Scanning Tunneling Microscopy/Scanning Tunneling Spectroscopy (STM/STS) I would like to mention that, it is not always necessary to have a conductive substrate. While grounding the sample can be achieved by improving the device architecture (Nano Letter 2016, 16, 4831-4837; ACS Nano 2021, 15, 2849-2857; Nature Communications 2023, 14, 5548). Furthermore, polymer contamination can also be re-moved using the AFM ironing procedure (Applied Physics Letters 2012, 100, 073110; Nature Communica-tions 2023, 14, 5548), which is now a standard method. Although these techniques require optimization and additional time to obtain results, they present a potential future perspective for the present work.

Reviewer #2

(Remarks to the Author)

Kim and coworkers have answered all the commentaries in quite a satisfying manner and have also improved the manuscript by adding a none negligible quantity of new experimental data. In this current from I recommend the manuscript be considered for publication.

Reviewer #3

(Remarks to the Author)

My concerns have been addressed in the revised manuscript with further measurements and discussions. I recommend this manuscript be considered for publication in NC.

Author Rebuttal letter:

Point by Point Replies to Reviewer's Comments

Reviewer #1

In the revised manuscript, the authors provide new experimental evidence and more detailed interpretations to address my previous comments. In general, I agree with most of the explanations provided by the authors. Based on the current revisions, I believe the manuscript is suitable for publication after addressing the following issues:

Reply: We appreciate Reviewer #1's positive comments and appreciation for our revisions.

I still have concerns regarding the interpretation of the PL spectra (previous comment 6).

1. It is well known that TMDs exhibit complex excitonic states, such as neutral excitons, trions, biexcitons, and localized states. The authors need to be very careful in assigning these peaks. Specifically, in the blue curve, it is unclear why the WSe₂ neutral exciton (which was not shown) and trion (1.74 eV) peak energies are so different from the other data (Figure 3b, Figure S12), where the neutral exciton and trion positions are approximately 1.74 eV and 1.70 eV, respectively. If the authors claim that at low temperatures the energy is upshifted (even though for MoSe₂ it does not shift much as we can see in Figure 3b and Figure S13b), then they must show the presence of the neutral exciton, which would be approximately 30 meV higher than the trion (Nature Materials 2013,12,207-211; ACS Nano 2021, 15, 2849-2857). In contrast, the MoSe₂ peaks are distinctly visible and consistent. Please address this discrepancy with a more thorough explanation to support the peak assignment.

Reply: We thank the reviewer for their careful observation and pointing this discrepancy to us.

In the supplementary information, we identified errors in assigning the WSe₂ peak in Figure S13. The primary peak labeled with XT is the neutral excitons and the trion (XT) is the small shoulder at 727 nm (~1.715 eV). We have replaced Supplementary Figure S13 with the following revised figure.

[Image redacted]

Figure S13. Temperature-dependent PL spectra of reference MoSe₂ and WSe₂ monolayers.

(a) PL spectra of the WSe₂ (top, blue) and MoSe₂ monolayers (bottom, red) measured at 80 K (solid) and room temperature (blurry dot) using a 633 nm-laser excitation in a Linkam stage with a liquid nitrogen supply. (b) Cryogenic PL spectra of WSe₂ (top, blue) and MoSe₂ monolayers (bottom, red) measured at 1.6 K using a 640 nm-laser excitation in a cryostat. For comparison, the cryogenic PL spectrum (middle, black) on the 1-min MoSe₂ QDs@WSe₂ heterostructures was added, shown in Figure 4b. The emission peaks in the top and bottom spectra correspond to various excitonic species (neutral excitons, trions, and defect states) in MoSe₂ and WSe₂ monolayers, exhibiting broader FWHM values due to homogeneous broadening mechanisms like many-body effects, phonon interactions, and scattering processes within the materials.

2. Why is each individual layer's FWHM quite broad even at 1.6 K compared to MoSe₂QDs@WSe₂ in Figure S13? Please provide a comprehensive analysis to clarify this observation.

Reply: Thank you for your question. The emission peaks in Figure S13B, middle spectrum (MoSe₂QDs@WSe₂), represent the quantum emission states which are characterized by their inherently narrow linewidths as quantum-confined structures. In contrast, the emission peaks shown in Figure S13B, top and bottom spectra, correspond to various excitonic species, such as neutral excitons, trions, and defects states, for both MoSe₂ and WSe₂ monolayers. These excitonic species exhibit broader FWHM values due to homogeneous broadening mechanisms including many-body effects, phonon interactions, and scattering processes within the materials.

We added the following sentence in the caption of Supplementary Figure S13 to account for the above explanation.

âThe emission peaks in the top and bottom spectra correspond to various excitonic species (neutral excitons, trions, and defect states) in MoSe₂ and WSe₂ monolayers, exhibiting broader

2

FWHM values due to homogeneous broadening mechanisms like many-body effects, phonon interactions, and scattering processes within the materials.â

Regarding the use of Scanning Tunneling Microscopy/Scanning Tunneling Spectroscopy (STM/STS) I would like to mention that, it is not always necessary to have a conductive substrate. While grounding the sample can be achieved by improving the device architecture (Nano Letter 2016, 16, 4831-4837; ACS Nano 2021, 15, 2849-2857; Nature Communications 2023, 14, 5548). Furthermore, polymer contamination can also be removed using the AFM ironing procedure (Applied Physics Letters 2012, 100, 073110; Nature Communications 2023, 14, 5548), which is now a standard method. Although these techniques require optimization and additional time to obtain results, they present a potential future perspective for the present work.

Reply: Thank you for the insightful suggestion. We agree that the proposed experiment would be valuable. However, as the reviewer mentions, performing STM/STS on insulating substrates requires optimization and additional time to obtain conclusive results. We believe that the TEM, near-field, and cryogenic optical analysis presented in our manuscript adequately support our claims. Although the STM/STS study is currently beyond our capabilities, we plan to conduct further experiments with our samples in the future to enhance their applicability. To the reviewer's point however, we have included a statement with several reference papers in the revised manuscript (page 10).

âFurthermore, a scanning tunneling microscopy/spectroscopy (STM/STS) study to provide more detailed information on the interface and defect states would be precious for this material system and present an opportunity for future research. Such a study, while challenging on an insulating growth substrate, could be done with improved device/sample architectures³¹⁻³³ and sample preparation techniques that minimize or eliminate polymer contaminations³³. Nonetheless, our TEM results (Figure 1, Supplementary Figures S1 and S2) provide evidence of the abruptness of the quantum heterostructure at the atomic level.â

31. Hill HM, Rigosi AF, Rim KT, Flynn GW, Heinz TF. Band Alignment in MoS₂/WS₂ Transition Metal Dichalcogenide Heterostructures Probed by Scanning Tunneling Microscopy and Spectroscopy. Nano Letters 2016, 16(8): 4831-4837.

32. Sebait R, Biswas C, Song B, Seo C, Lee YH. Identifying Defect-Induced Trion in Monolayer WS₂ via Carrier Screening Engineering. ACS Nano 2021, 15(2): 2849-2857.

33. Sebait R, Rosati R, Yun SJ, Dhakal KP, Brem S, Biswas C, et al. Sequential order dependent dark-exciton modulation in bi-layered TMD heterostructure. Nature Communications 2023, 14(1): 5548.

Reviewer #2:

Kim and coworkers have answered all the commentaries in quite a satisfying manner and have also improved the manuscript by adding a none negligible quantity of new experimental data. In this current from I recommend the manuscript be considered for publication.

Reply: We appreciate Reviewer #2 reading our work once again and recommending our work for publication.

3

Reviewer #3:

My concerns have been addressed in the revised manuscript with further measurements and discussions. I recommend this manuscript be considered for publication in NC.

Reply: We appreciate Reviewer #3 reading our work once again and recommending our work for publication.

Version 2:

Author Rebuttal letter:

Point by Point Replies to Editor's Comments

Reviewer #1

In the revised manuscript, the authors provide new experimental evidence and more detailed interpretations to address my previous comments. In general, I agree with most of the explanations provided by the authors. Based on the current revisions, I believe the manuscript is suitable for publication after addressing the following issues:

Reply: We appreciate Reviewer #1's positive comments and appreciation for our revisions.

I still have concerns regarding the interpretation of the PL spectra (previous comment 6).

1. It is well known that TMDs exhibit complex excitonic states, such as neutral excitons, trions, biexcitons, and localized states. The authors need to be very careful in assigning these peaks. Specifically, in the blue curve, it is unclear why the WSe₂ neutral exciton (which was not shown) and trion (1.74 eV) peak energies are so different from the other data (Figure 3b, Figure S12), where the neutral exciton and trion positions are approximately 1.74 eV and 1.70 eV, respectively. If the authors claim that at low temperatures the energy is upshifted (even though for MoSe₂ it does not shift much as we can see in Figure 3b and Figure S13b), then they must show the presence of the neutral exciton, which would be approximately 30 meV higher than the trion (Nature Materials 2013, 12, 207-211; ACS Nano 2021, 15, 2849-2857). In contrast, the MoSe₂ peaks are distinctly visible and consistent. Please address this discrepancy with a more thorough explanation to support the peak assignment.

Reply: We thank the reviewer for their careful observation and pointing this discrepancy to us. In the supplementary information, we identified errors in assigning the WSe₂ peak in Figure S13. The primary peak labeled with XT is the neutral exciton and the trion (XT) is the small shoulder at 727 nm (~1.715 eV). We have replaced Supplementary Figure S13 with the following revised figure.

[Image redacted]

Figure S13. Temperature-dependent PL spectra of reference MoSe₂ and WSe₂ monolayers. (a) PL spectra of the WSe₂ (top, blue) and MoSe₂ monolayers (bottom, red) measured at 80 K (solid) and room temperature (blurry dot) using a 633 nm-laser excitation in a Linkam stage with a liquid nitrogen supply. (b) Cryogenic PL spectra of WSe₂ (top, blue) and MoSe₂ monolayers (bottom, red) measured at 1.6 K using a 640 nm-laser excitation in a cryostat. For comparison, the cryogenic PL spectrum (middle, black) on the 1-min MoSe₂ QDs@WSe₂ heterostructures was added, shown in Figure 4b. The emission peaks in the top and bottom spectra correspond to various excitonic species (neutral excitons, trions, and defect states) in MoSe₂ and WSe₂ monolayers, exhibiting broader FWHM values due to homogeneous broadening mechanisms like many-body effects, phonon interactions, and scattering processes within the materials.

2. Why is each individual layer's FWHM quite broad even at 1.6 K compared to MoSe₂QDs@WSe₂ in Figure S13? Please provide a comprehensive analysis to clarify this observation.

Reply: Thank you for your question. The emission peaks in Figure S13B, middle spectrum (MoSe₂QDs@WSe₂), represent the quantum emission states which are characterized by their inherently narrow linewidths as quantum-confined structures. In contrast, the emission peaks shown in Figure S13B, top and bottom spectra, correspond to various excitonic species, such as neutral excitons, trions, and defect states, for both MoSe₂ and WSe₂ monolayers. These excitonic species exhibit broader FWHM values due to homogeneous broadening mechanisms including many-body effects, phonon interactions, and scattering processes within the materials.

We added the following sentence in the caption of Supplementary Figure S13 to account for the above explanation.

âThe emission peaks in the top and bottom spectra correspond to various excitonic species (neutral excitons, trions, and defect states) in MoSe₂ and WSe₂ monolayers, exhibiting broader

FWHM values due to homogeneous broadening mechanisms like many-body effects, phonon interactions, and scattering processes within the materials.^â

Regarding the use of Scanning Tunneling Microscopy/Scanning Tunneling Spectroscopy (STM/STS) I would like to mention that, it is not always necessary to have a conductive substrate. While grounding the sample can be achieved by improving the device architecture (Nano Letter 2016, 16, 4831-4837; ACS Nano 2021, 15, 2849-2857; Nature Communications 2023, 14, 5548). Furthermore, polymer contamination can also be removed using the AFM ironing procedure (Applied Physics Letters 2012, 100, 073110; Nature Communications 2023, 14, 5548), which is now a standard method. Although these techniques require optimization and additional time to obtain results, they present a potential future perspective for the present work.

Reply: Thank you for the insightful suggestion. We agree that the proposed experiment would be valuable. However, as the reviewer mentions, performing STM/STS on insulating substrates requires optimization and additional time to obtain conclusive results. We believe that the TEM, near-field, and cryogenic optical analysis presented in our manuscript adequately support our claims. Although the STM/STS study is currently beyond our capabilities, we plan to conduct further experiments with our samples in the future to enhance their applicability. To the reviewer's point however, we have included a statement with several reference papers in the revised manuscript (page 10).

âFurthermore, a scanning tunneling microscopy/spectroscopy (STM/STS) study to provide more detailed information on the interface and defect states would be precious for this material system and present an opportunity for future research. Such a study, while challenging on an insulating growth substrate, could be done with improved device/sample architectures³¹⁻³³ and sample preparation techniques that minimize or eliminate polymer contaminations³³.

Nonetheless, our TEM results (Figure 1, Supplementary Figures S1 and S2) provide evidence of the abruptness of the quantum heterostructure at the atomic level.^â

31. Hill HM, Rigosi AF, Rim KT, Flynn GW, Heinz TF. Band Alignment in MoS₂/WS₂ Transition Metal Dichalcogenide Heterostructures Probed by Scanning Tunneling Microscopy and Spectroscopy. Nano Letters 2016, 16(8): 4831-4837.

32. Sebait R, Biswas C, Song B, Seo C, Lee YH. Identifying Defect-Induced Trion in Monolayer WS₂ via Carrier Screening Engineering. ACS Nano 2021, 15(2): 2849-2857.

33. Sebait R, Rosati R, Yun SJ, Dhakal KP, Brem S, Biswas C, et al. Sequential order dependent dark-exciton modulation in bi-layered TMD heterostructure. Nature Communications 2023, 14(1): 5548.
